# The somatic mutation landscape of normal gastric epithelium

Tim H. H. Coorens[1,2 ✉], Grace Collord[1,3], Hyungchul Jung[1], Yichen Wang[1], Luiza Moore[1], Yvette Hooks[1], Krishnaa Mahbubani[4,5], Simon Y. K. Law[6], Helen H. N. Yan[7,8], Siu Tsan Yuen[7,9], Kourosh Saeb-Parsy[4,5], Peter J. Campbell[1,10,11], Iñigo Martincorena[1], Suet Yi Leung[7,8 ✉] & Michael R. Stratton[1 ✉]

The landscapes of somatic mutation in normal cells inform us about the processes of mutation and selection operative throughout life, providing insight into normal ageing and the earliest stages of cancer development[1]. Here, by whole-genome sequencing of 238 microdissections[2] from 30 individuals, including 18 with gastric cancer, we elucidate the developmental trajectories of normal and malignant gastric epithelium. We find that gastric glands are units of monoclonal cell populations that accrue roughly 28 somatic single-nucleotide variants per year, predominantly attributable to endogenous mutational processes. In individuals with gastric cancer, metaplastic glands often show elevated mutation burdens due to acceleration of mutational processes linked to proliferation and oxidative damage. Unusually for normal cells, gastric epithelial cells often carry recurrent trisomies of specific chromosomes, which are highly enriched in a subset of individuals. Surveying 829 polyclonal gastric microbiopsies by targeted sequencing, we find somatic 'driver' mutations in a distinctive repertoire of known cancer genes, including *ARID1A*, *ARID1B*, *ARID2*, *CTNNB1* and *KDM6A*. The prevalence of mutant clones increases with age to occupy roughly 8% of the gastric epithelial lining by age 60 years and is significantly increased by the presence of severe chronic inflammation. Our findings provide insights into intrinsic and extrinsic influences on somatic evolution in the gastric epithelium in healthy, precancerous and malignant states.

Over the course of a lifetime, cells in the human body acquire somatic mutations, thus generating genetic diversity and enabling natural selection in tissues. Until recently, understanding of the somatic mutation landscape of normal cells has been limited compared to that of cancer cells. However, new DNA sequencing approaches have enabled exploration of normal somatic cell genomes, elucidation of cell lineages, estimation of mutation rates, assessment of underlying mutational processes and detection of clones carrying mutated genes conferring selective growth advantage[1,3,4]. These mutation landscapes provide insights into somatic evolution in normal tissues during an individual's lifetime and into the earliest stages of cancer development[5–13].

The gastrointestinal tract constitutes four main segments—the oesophagus, stomach, small intestine and large intestine—which serially process ingested food materials and interface with very different types of luminal content. The somatic mutation landscapes of normal epithelial cells lining the oesophagus[5], small intestine[7] and large intestine[6] have recently been characterized. The stomach comprises several anatomically and histologically distinct regions, including the cardia, fundus, body, lesser and greater curvatures, antrum and pylorus. The epithelial lining of the stomach is composed of branched glands producing hydrochloric acid, digestive enzymes and hormones.

Gastric cancer is the fifth commonest cancer diagnosis globally and the third leading cause of cancer-related death[14]. Its incidence varies geographically and is highest in East Asia and South America[15]. Known risk factors include infection with *Helicobacter pylori* and Epstein–Barr virus, alcohol use, tobacco, obesity and diet[14–16]. Cancer risks and the influence of different risk factors differ profoundly between anatomical domains of the stomach, with the highest risks in the antrum in regions with high incidence[16] and in the cardia in regions with low incidence[15,16]. The epidemiology of gastric cancer indicates that many extrinsic factors, through exposures and chronic inflammation, influence somatic mutagenesis in the stomach. Chronic inflammation can lead to metaplasia, a remodelling of the gastric epithelium to resemble intestinal epithelium, which is thought of as a precursor to overt cancer[17].

Here we investigate the somatic genetic diversity in gastric epithelium from donors with and without gastric cancer and begin to shed light on the transition between normal age-related somatic evolution and malignancy.

[1]Wellcome Sanger Institute, Hinxton, UK. [2]Broad Institute of MIT and Harvard, Cambridge, MA, USA. [3]University College London Hospital, London, UK. [4]Department of Surgery, University of Cambridge, Cambridge, UK. [5]Cambridge NIHR Biomedical Research Centre, Cambridge Biomedical Campus, Cambridge, UK. [6]Department of Surgery, School of Clinical Medicine, The University of Hong Kong, Queen Mary Hospital, Pokfulam, China. [7]Department of Pathology, School of Clinical Medicine, The University of Hong Kong, Queen Mary Hospital, Pokfulam, China. [8]Centre for Oncology and Immunology, Hong Kong Science Park, Hong Kong, China. [9]Department of Pathology, St. Paul's Hospital, No. 2, Eastern Hospital Road, Causeway Bay, China. [10]Wellcome-MRC Cambridge Stem Cell Institute, Cambridge Biomedical Campus, Cambridge, UK. [11]Department of Haematology, University of Cambridge, Cambridge, UK. ✉e-mail: tcoorens@broadinstitute.org; suetyi@hku.hk; mrs@sanger.ac.uk

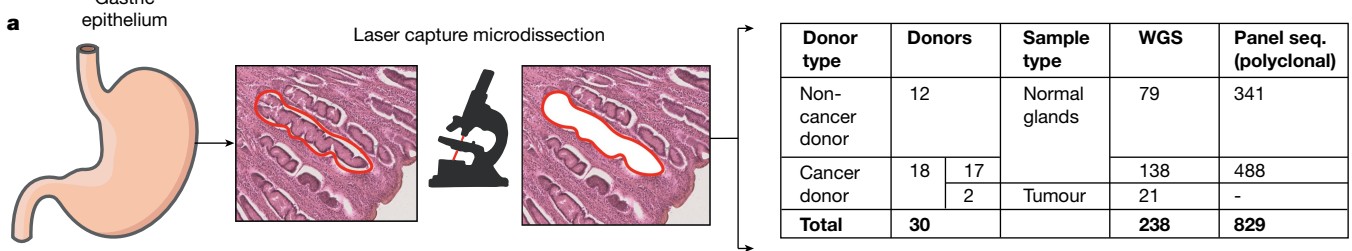

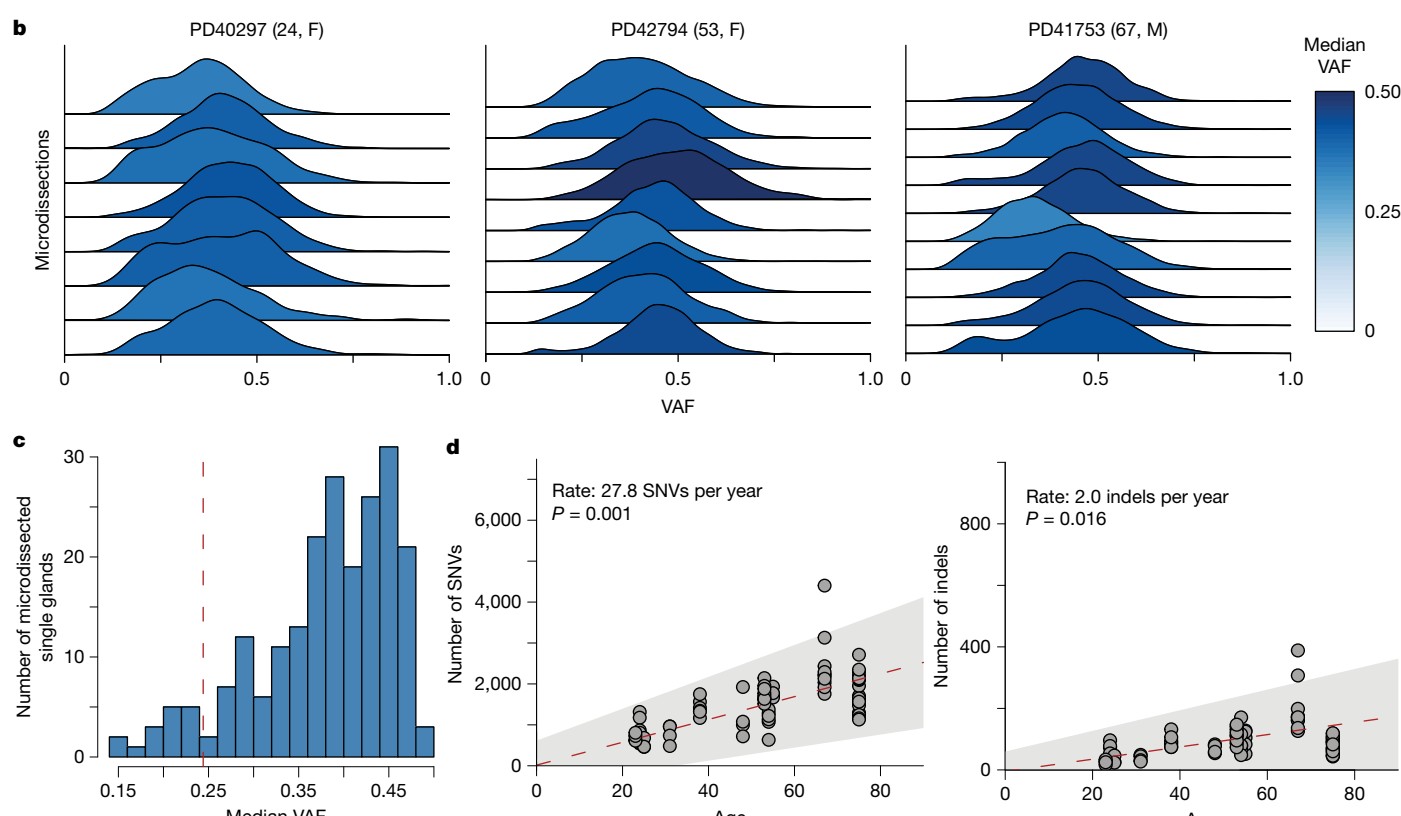

**Fig. 1 | Clonality and mutation rates. a**, Overview of the study. Gastric glands were sampled from 30 donors. Two hundred seventeen microdissected individual glands from normal, inflamed and/or metaplastic stomach tissue and 21 gastric cancer gland microdissections were whole-genome sequenced, and a further 829 microdissections, each comprising several adjacent glands, were subjected to deep targeted gene sequencing (seq.). **b**, VAF distributions of somatic mutations in gastric gland microdissections for three donors, coloured by the median VAF. **c**, Histogram of median VAF for all WGS microdissections of single gastric glands (*n* = 217). **d**, Number of SNVs and indels plotted against the age of the donors for gastric glands from non-cancer donors (*n* = 79). The red dashed line indicates the maximum-likelihood estimated age and SNV mutation burden relation estimated from a mixed-effects model, with the grey shaded area indicating the 95% confidence interval. *P* values in **d** are obtained through a two-sided ANOVA test. Credit: stomach outline in **a** adapted from Servier Medical Art (https://smart.servier.com/) under a CC BY 4.0 International licence. Outline of laser capture microdissection in **a** adapted from ref. 8, Springer Nature Limited. M, male; F, female.

## Mutation rates of gastric epithelium

The cohort consists of 30 individuals, 18 with gastric cancer and 12 with no gastric pathology (referred to subsequently as 'cancer donors' and 'non-cancer donors', respectively), from Hong Kong, the United States and the United Kingdom (Supplementary Table 1). Donors from Hong Kong were tested for infection with *H. pylori*. Two hundred seventeen gastric glands were microdissected and individually whole-genome sequenced to 23-fold median coverage, along with 21 neoplastic glands from the gastric cancers of two individuals (Fig. 1a and Supplementary Table 2). In addition, we subjected a further 829 microdissections of individual or clustered gastric glands to targeted sequencing of known cancer genes. All classes of somatic mutation were called by standard approaches (Methods).

Unlike the straight tubular glands of the intestine, gastric glands consist of tubules that branch towards the base, with a potential contribution of self-renewing stem cells. Previous research from X-inactivation studies has shown that gastric glands start as polyclonal units and drift to monoclonality, although polyclonal glands may persist throughout life[18]. The clonal composition of gastric glands can be estimated from the variant allele fractions (VAFs) of somatic single-nucleotide variants (SNVs) and small insertions and deletions (indels). The median VAF per microdissection generally exceeded 0.25 (Fig. 1b,c), which—allowing for a degree of stromal contamination—confirms the notion that most glands are dominated by the progeny of a single stem cell, a clone that takes up more than half of all cells. In 8% of microdissections (17 of 217), the median VAF was below 0.25 or had evidence of several clones co-existing, indicating a continued presence of several stem cell niches and hierarchies more complicated than those observed in intestinal crypts[6].

The total burden of somatic SNVs in normal glands from the 12 individuals without gastric cancer increased linearly with age (Fig. 1d), such

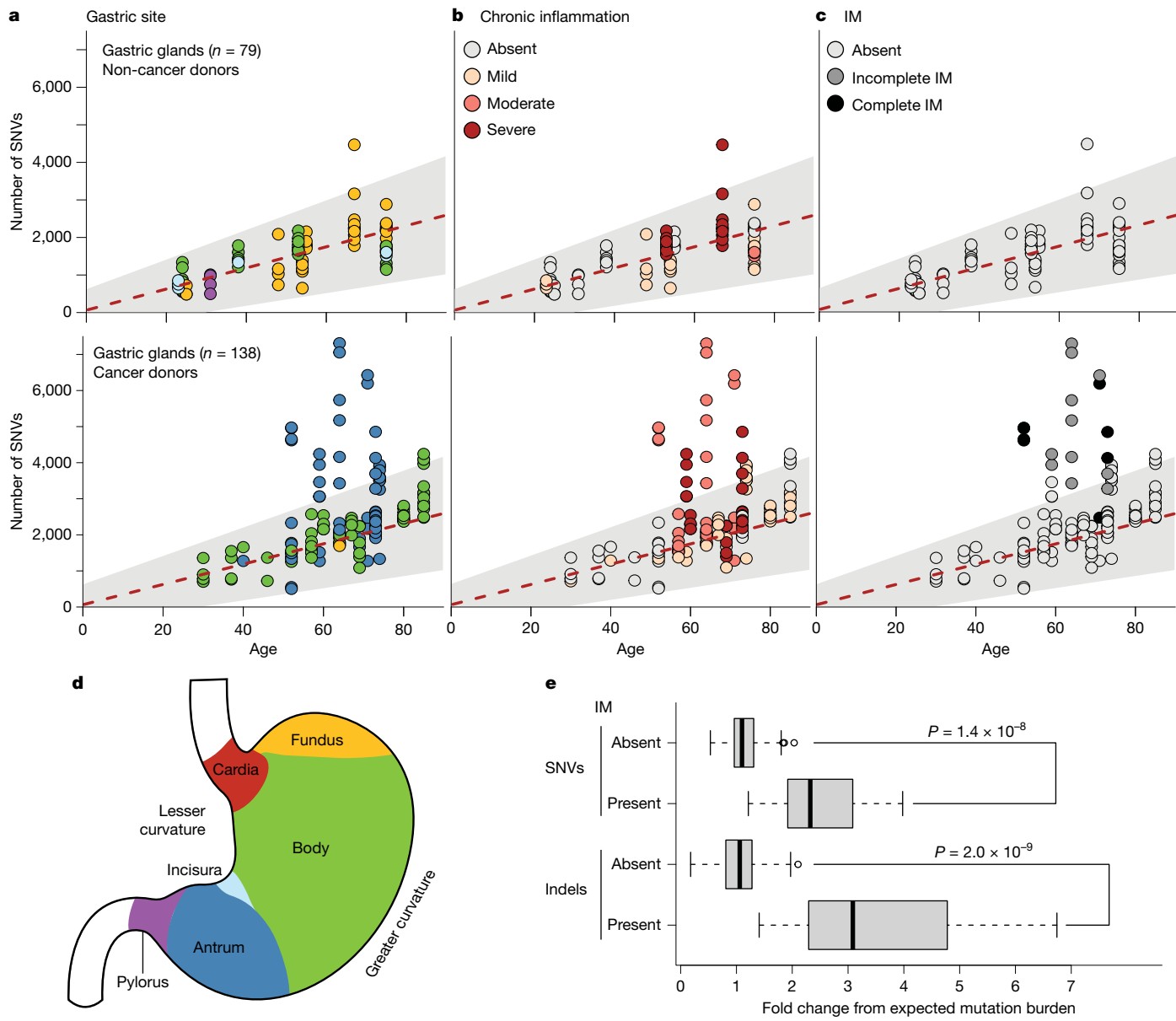

**Fig. 2 | Gastric segments, chronic inflammation and metaplasia.**
**a**–**c**, Scatterplots showing numbers of SNVs plotted against the age of the donors for gastric glands (**a**) in non-cancer donors (top row) and cancer donors (bottom row), coloured by the section of the stomach sampled, as shown in the schematic in **d**, chronic inflammation status (**b**) and presence of IM (**c**). The red dashed line indicates the maximum-likelihood age and SNV mutation burden relation estimated from a mixed-effects model in gastric glands from non-cancer donors (Fig. 1d), with the grey shaded area indicating the 95% confidence interval. **d**, Schematic of gastric anatomy. **e**, Boxplots of ratios between observed and age-related expected SNV and indel mutation burden (fold change) in gland microdissections from cancer donors ($n = 138$) with and without IM. The central line, box and whiskers represent the median, interquartile range (IQR) from first to third quartiles and 1.5× IQR, respectively. $P$ values obtained from a two-sided Wilcoxon rank-sum test. Credit: stomach outline in **d** adapted from Servier Medical Art (https://smart.servier.com/) under a CC BY 4.0 International licence.

that their stem cell progenitors accrue about 27.8 SNVs (95% confidence interval: 16.2–39.4) and 2.0 indels (95% confidence interval: 0.74–3.28) per year. The clonality (that is, median VAF) of the gland did not correlate with mutation burdens after correcting for sensitivity of variant calling (Extended Data Fig. 1a), which depends on sequencing depth and observed VAF distributions.

The burdens of SNVs and indels in microdissections of gastric glands from donors with gastric cancer largely followed the age-related increase observed in individuals without gastric cancer (Fig. 2a–c), with the notable exception of glands with intestinal metaplasia (IM) ($n = 19$, $P = 1 × 10^{-42}$ for SNVs and $P = 1.8 × 10^{-49}$ for indels, analysis of variance (ANOVA) test). Although chronic inflammation was widespread in both non-cancer and cancer donors, IM, both complete and

incomplete, was exclusive to the antrum of individuals with gastric cancer in our cohort. On average, the mutation burdens in metaplastic glands were increased 2.8-fold and 4.4-fold for SNVs and indels, respectively, compared to the age-expected burdens (Fig. 2e). Metaplastic glands exhibited overall higher median VAFs per microdissection compared to non-metaplastic glands ($P = 0.006$, Wilcoxon rank-sum test; Extended Data Fig. 1b) and were closely related phylogenetically when anatomically close to each other (Extended Data Fig. 2a,c), indicating that metaplastic clones locally expand. However, metaplastic glands from the same donor at a distance from each other were phylogenetically unrelated beyond early development (Extended Data Fig. 2a,b), indicating a wider 'field effect' of metaplasia induction.

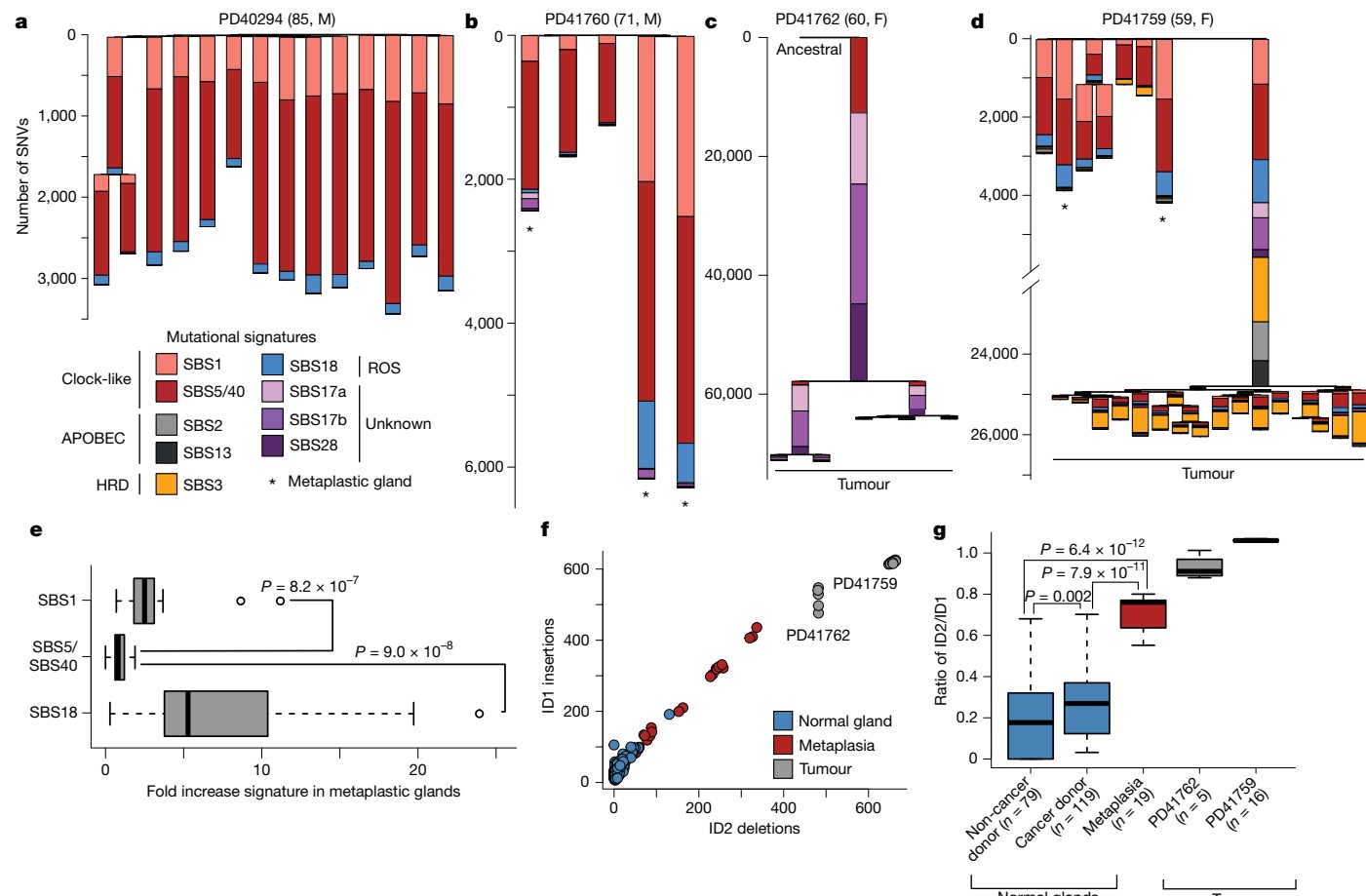

**Fig. 3 | Mutational signatures. a–d**, Phylogenetic trees of gastric glands from four different donors with gastric cancer. **a**, PD40294 (non-metaplastic glands only). **b**, PD41760 (metaplastic and non-metaplastic glands). **c**, PD41762 (tumour only). **d**, PD41759 (tumour, metaplastic and non-metaplastic glands). Mutational signature proportions are overlaid on each branch and different signatures are indicated by different colours (see legend). Branch length represents the number of SNVs on each branch. Note the broken $y$ axis in **d**. Asterisks denote glands with IM and microdissections of glands from cancers are indicated in the phylogenies. **e**, Fold increase of SBS1, SBS5/40 and SBS18 in the 19 microdissections from metaplastic glands estimated by comparing the observed mutation burdens of these signatures compared to the expected burdens. $P$ values are obtained through two-sided Wilcoxon rank-sum tests. The central line, box and whiskers represent the median, IQR from first to third quartiles and 1.5× IQR, respectively. **f**, Number of ID1 insertions versus ID2 deletions across samples, both associated with polymerase slippage. **g**, The ratio of ID2 deletions to ID1 insertions for normal glands, metaplastic glands and tumour samples. The central line, box and whiskers represent the median, IQR range from first to third quartiles and 1.5× IQR, respectively. $P$ values are obtained from two-sided Wilcoxon rank-sum tests. ROS, reactive oxygen species; HRD, homologous repair deficiency.

Annotated current or previous *H. pylori* status, known in only a minority of donors, did not significantly affect SNV burdens ($P = 0.74$, ANOVA test; Extended Data Fig. 1c). However, the possibility of undetected infections affecting mutation rates precludes a definitive conclusion on this relationship. The SNV and indel burdens of microdissected glands from gastric cancers were substantially elevated compared to the mutation loads observed in normal or metaplastic gastric glands (Extended Data Fig. 1d,e). From whole-genome sequencing (WGS), we estimate that telomeres shorten by an average of 38 bases per year (95% confidence interval: 25–53) in gastric glands, with a significant shortening of telomeres by a mean of 570 bases in the presence of moderate or severe chronic inflammation ($P = 6 \times 10^{-5}$, ANOVA test; Extended Data Fig. 1f). Beyond the effect of chronic inflammation, metaplasia did not further reduce telomere length ($P = 0.11$, ANOVA test).

## Mutational signatures and processes

Mutational signatures are the patterns of mutation imprinted on the genome by the activity of specific mutational processes. Their contributions to the somatic mutations found in individual samples can be established using mathematical approaches for their deconvolution and attribution. More than 70 single-base substitution (SBS) reference signatures have been reported in cancer and normal cells (https://cancer.sanger.ac.uk/signatures).

Using the whole-genome somatic mutation catalogues of all gastric glands, nine mutational signatures were extracted (Fig. 3a–d and Extended Data Figs. 3–4), all of which have been previously reported[19,20] (Methods): SBS1, due to spontaneous deamination of 5-methylcytosine; SBS2 and SBS13, due to activity of APOBEC cytidine deaminases and identified only in the tumour of PD41759 (Fig. 3d); SBS3, due to homologous repair deficiency and identified only in PD41759 (Fig. 3d); SBS5/SBS40, of unknown aetiologies but thought to be of intrinsic origin; SBS17a and SBS17b, of unknown aetiologies but sometimes associated with exposure to the chemotherapeutic agent 5-fluorouracil or oxidation of the free nucleotide pool[21]; SBS18, due to DNA damage by reactive oxygen species; and SBS28, of unknown aetiology.

Most SNVs in normal gastric glands were explained by SBS1, SBS5/40 and SBS18 (Fig. 3a,b), which are detected in all gastric glands. SBS1 and SBS5/SBS40 are ubiquitous in human cancers[19] and in normal cells[1]. SBS18 is observed in many normal cell types, particularly those with

high cell-division rates[6,8,11]. In normal, non-metaplastic gastric glands, the mutation burdens attributable to SBS1, SBS5/40 and SBS18 linearly correlated with age (Extended Data Fig. 5). The higher-than-expected SNV mutation burdens found in metaplastic gastric glands were due to increased mutation burdens of SBS1 (approximately 3-fold) and SBS18 (approximately 8-fold), but not SBS5/40 (approximately 1-fold) (Fig. 3e).

Signature analysis of indels showed that gastric glands ubiquitously exhibited ID1 and ID2—single-base insertions and deletions, respectively—at homopolymer runs of T/A linked to polymerase slippage and ID5 and ID9, both characterized by deletions at homopolymers but of unknown aetiology (Extended Data Fig. 6a,b). The excess of indels observed in metaplastic glands was primarily composed of ID1 and ID2 (Fig. 3f and Extended Data Fig. 6c) but the ratio of ID2 to ID1 was significantly elevated in metaplastic glands compared to other non-cancer glands and was more akin to that of gastric cancer samples (Fig. 3g).

A small subset of normal gastric glands in individuals with gastric cancer also exhibited modest burdens of SBS17a and SBS17b (Fig. 3b). Substantial SBS17a and SBS17b mutation loads are common in oesophageal adenocarcinoma[22] and its precursor lesion, Barrett's oesophagus[23], as well as in gastric adenocarcinoma[19], as illustrated by the two cancers sequenced here (Fig. 3c,d). By contrast, these mutational signatures were rarely observed in normal stomach, indicating that the mutational processes underlying SBS17a and SBS17b are primarily features of neoplastic cells. Nevertheless, SBS17a and SBS17b have been identified in only one other normal cell type (B lymphocytes[9]), and thus there seems to be a particular propensity of gastric epithelial cells to generate them and/or the existence of gastric microenvironmental factors to induce them.

The pattern of mutational signatures in glands dissected from the two gastric cancers was markedly different from that in normal glands. Although still exhibiting contributions from the aforementioned ubiquitous SNV and indels signatures, the cancers showed large contributions from SBS17a and SBS17b, as well as SBS3 and ID6 (homologous repair deficiency), SBS2 and SBS13 (APOBEC activity) (Fig. 3c,d) and ID14 (unknown aetiology), consistent with previously reported series[19].

## Recurrent trisomies in gastric glands

Somatic copy-number variants (CNVs) and structural variants were observed in a minority of normal gastric glands (73 of 217), but nevertheless at considerably higher prevalence than other normal human cell types studied thus far[1,6,11] (Fig. 4a,b). Moreover, the CNVs in gastric epithelium exhibited a highly distinctive pattern. Intrachromosomal CNVs and structural variants were predominantly deletions and many involved well-known fragile sites in FHIT, PTPRD and MACROD2 (refs. 24,25). Although these have been reported in gastric cancer[24], it is unclear whether the observed events are due to high rates of mutagenesis in the locus or positive selection in this data. Chromosome arm-level events were all copy-number-neutral loss of heterozygosity (cnn-LOH), whereas whole-chromosome events exclusively comprised somatic trisomies, mostly of chromosomes 13 and 20. Remarkably, trisomies were concentrated in a subset of individuals and had often arisen independently and several times in the same individual (Fig. 4b) at similar ages (Extended Data Fig. 7). This independent origin of trisomies was inferred from the phylogenetic tree topology (Fig. 4c,d) and further corroborated by the presence of different duplicated SNVs on the trisomic chromosomes between samples and the duplication of different parental copies in different gastric glands.

For example, in a 64-year-old man with gastric cancer, PD40293, six out of 12 gastric glands analysed exhibited chromosome 20 trisomy, three exhibited chromosome 13 trisomy, and one exhibited cnn-LOH of 17q (Fig. 4c). Eight glands showed just a single CNV, and one showed both trisomy 13 and trisomy 20. Thus, nine of 12 glands showed CNVs, indicating that a substantial proportion of the gastric epithelium had

been colonized by cells with CNVs. The results show five independent duplications of chromosome 20, three of one parental copy and two of the other, and two independent duplications of chromosome 13 among the 12 sampled glands. Using the relative proportions of duplicated and non-duplicated SNVs, we estimate that all five trisomies of chromosome 20 occurred relatively early in life, around or before age 12, the two chromosome 13 duplications around or before age 22 and the cnn-LOH of 17q around or before age 35. Analyses of gastric cancer genomes indicate trisomy 20 is a predominantly early event[26], corroborating the time scales estimated here.

By interrogating SNPs on chromosome 20 from panel sequencing data in PD40293 and using the phasing information from WGS data (Extended Data Fig. 8), we were further able to identify microdissections with trisomy 20 (23 out of 65), confirming the widespread nature of this somatic change. In this donor, occurences of trisomy 20 in both panel and whole-genome sequenced samples were strongly enriched in the fundus (21 of 34) and body (6 of 17) over the antrum (2 of 24) ($P = 4.5 \times 10^{-5}$, Fisher's exact test).

The cause of this distinctive pattern of CNVs in gastric glands is uncertain. In PD41767, trisomies are detected in four of ten glands in one severely inflamed stomach biopsy of the antrum, but wholly absent from seven glands sampled from another section of the antrum, which was much less inflamed. Both age and the presence of metaplasia have a significant effect on the burden of intrachromosomal structural variants and CNVs ($P = 0.01$ and $P = 10^{-14}$, respectively, ANOVA test). Although the burden of trisomies is not significantly linearly associated with age ($P = 0.38$), metaplasia ($P = 0.84$) or whether the donor had gastric cancer ($P = 0.63$), there is a significant association with severe chronic inflammation ($P = 0.004$; all tests are ANOVA tests).

Our data indicate that rather than a continuous age-associated increase of whole-chromosome duplications, these trisomies were generated at a specific time during the lifespan of each individual and possibly confined to specific regions of the stomach. The process of acquisition or selection of these trisomies is not apparently linked with metaplasia or carcinogenesis, but there is possible a link with chronic inflammation. Although only one of the donors harbouring trisomies was known to be infected with H. pylori, the confinement in space and time, and the association with inflammation, are suggestive of the involvement of an exposure or infection with a pathogen.

## Driver mutations in gastric glands

To systematically identify genes under positive selection, we supplemented the whole-genome data from the 217 glands with targeted sequencing of 321 known cancer genes in a further 829 polyclonal microdissections (Methods). Seven mutated genes showed statistically significant ($q < 0.1$) evidence of positive selection (Fig. 5a–c): ARID1A, ARID1B and ARID2, subunits of the SWI/SNF chromatin remodelling complex; CTNNB1, a WNT signalling pathway transducer and cell adhesion molecule; KDM6A, a regulator of histone methylation; LIPF, encoding gastric lipase; and EEF1A1, a translation elongation factor. All these genes, with the exception of LIPF and EEF1A1, have been reported as frequently mutated in gastric cancer[27]. LIPF is a highly expressed gene in gastric epithelium and may be prone to accelerated mutagenesis, as previously reported in gastric cancer[28].

Intriguingly, the CTNNB1 mutations observed in normal gastric glands were predominantly protein-truncating nonsense substitutions and frameshifting indels scattered across the gene footprint that probably inactivate the encoded protein. This contrasts with the pattern of clustered hotspot missense CTNNB1 substitutions reported in gastric cancer and many other cancer types, characteristic of oncogene activation. The reason for this difference between normal and cancer cells in the pattern of CTNNB1 mutations is unclear but may again highlight the different selective advantage required for a normal cell to thrive in a normal tissue and for it to thrive as a cancer cell.

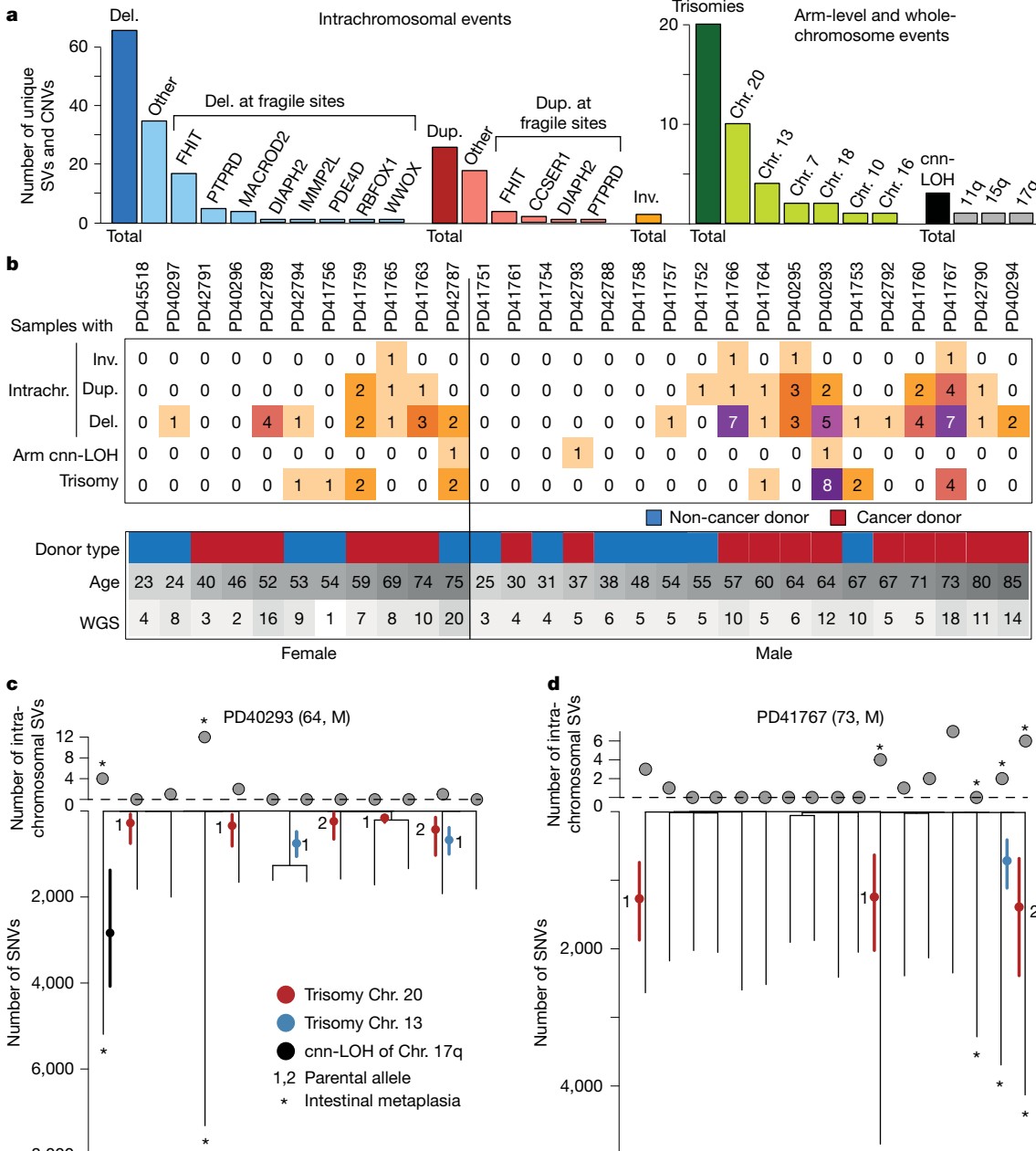

**Fig. 4 | CNVs and recurrent trisomies. a**, Overview of unique CNVs called across the WGS samples, split by the size of the event and further divided by specific site or chromosome. **b**, Heatmap of the number of microdissections with specific categories of structural variants (SVs) or CNVs per donor, along with donor cohort, age and number of WGS microdissections. **c,d**, Dot plots of the number of intrachromosomal structural variants (deletions, duplications or inversions) along phylogenetic trees of the gastric glands of two donors, PD40293 (**c**) and PD41767 (**d**), with recurrent gains of chromosome 13 or 20.

Branch length represents the number of SNVs on each branch. The timing of the gain is indicated by the red (chr. 20), blue (chr. 13) or black (chr. 17q cnn-LOH) dot, with solid, coloured lines representing the 95% Poisson confidence interval around this estimate based on the number of duplicated and non-duplicated SNVs in regions affected by CNVs (numbers listed in Supplementary Table 4). Numbers indicate the parental allele that is gained. Asterisks denote microdissections from metaplastic glands. Del., deletion; dup., duplication; inv., inversion.

In some instances, the incidence of driver mutations was highly confined to a specific location. For example, PD42790 harboured clones with three independent *CTNNB1* frameshift mutations (out of the four *CTNNB1* mutations detected in this donor) within millimetres, indicating a particularly strong local selective pressure in favour of these mutations (Fig. 5d).

Likely driver mutations in a further set of genes that did not reach formal significance levels were identified at known missense mutation hotspots in dominantly acting cancer genes (*BRAF*, *KRAS*), and, in addition, some protein-truncating mutations in tumour suppressor

genes (*APC*, *BCOR*) may also have conferred clonal growth advantage. However, mutations in *TP53* and *PIK3CA* were not observed in normal gastric glands despite being common in gastric adenocarcinomas and in some other normal cell types[27]. Driver mutations in the WGS data were largely confined to single microdissections (Extended Data Figs. 3 and 4), and although there was no enrichment of drivers in metaplastic glands (*P* = 1), glands with severe chronic inflammation were significantly enriched for drivers (*P* = 0.01, Fisher's exact tests). The proportion of the gastric epithelium, surveyed by both whole-genome and panel sequencing, colonized by mutant clones

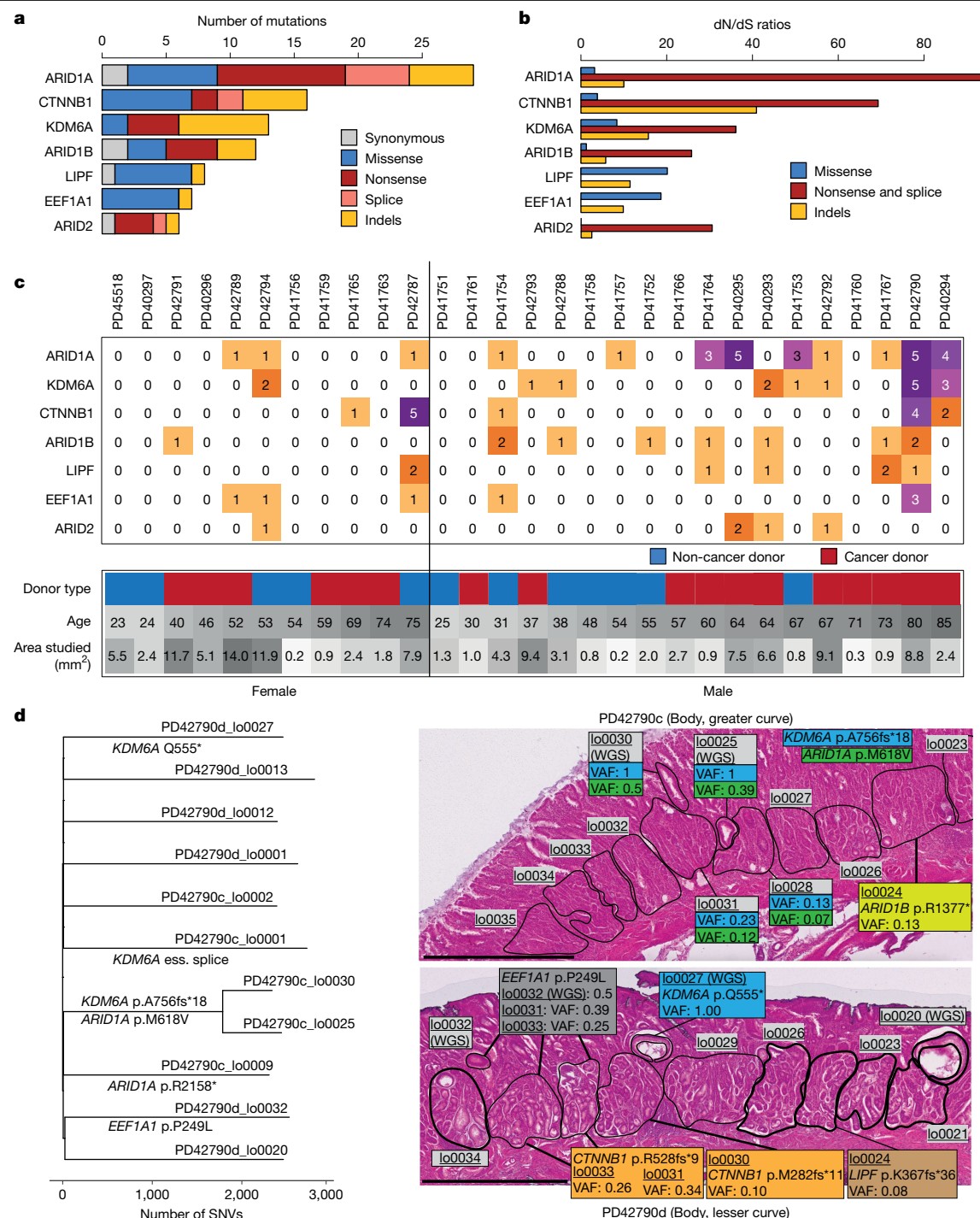

**Fig. 5 | Driver mutations. a**, Number of mutations, broken down by functional effect, for genes under significant positive selection. **b**, dN/dS ratios for genes under positive selection, broken down by type. **c**, Heatmap of distribution of specific driver mutations per donor. **d**, Phylogenetic tree of donor PD42790 (80, M) from the WGS data of individual glands, annotated with putative driver mutations, along with histology images from two regions overlaid with driver mutations and their VAF, of both whole-genome sequenced microdissections (indicated by WGS) and panel-sequenced clusters of microdissections. Scale bars, 1 mm.

likewise depended on age ($P = 0.002$) and severe chronic inflammation ($P = 0.001$), but not metaplasia ($P = 0.86$, ANOVA tests) (Extended Data Fig. 9a,b). Although the age dependency can be explained by the age-related increase in mutation burden, the effect of chronic inflammation on mutant proportion persists beyond differences in mutation burden (Extended Data Fig. 9c). On average, in 60-year-old individuals, about 7.8% of glands were colonized by clones with driver mutations.

## Discussion

The somatic mutation landscapes of the four primary segments of the gastrointestinal tract—the oesophagus[5,29], stomach, small intestine[7] and large intestine[6]—have now been surveyed to a first level of resolution, exhibiting illustrative similarities and differences. The lining epithelial cells of these segments interface with very different types of luminal content; these include air and the highly variable temperatures

of food in the oesophagus, the acid and sterile contents of the stomach reservoir, the neutral contents and limited microbiome of the small intestine, and the florid, diverse microbiome of the large intestine. Overall, however, the differences in somatic mutation rates and mutational signatures are modest.

All four gastrointestinal segments show a roughly constant mutation rate, ranging from about 30 SNVs per year in the oesophagus and stomach to about 50 SNVs per year in the small and large intestines, mostly generated by the biological processes underlying SBS1, SBS5/40 and SBS18, which are thought to be primarily of endogenous origin, albeit in different relative contributions. In addition to these ubiquitous mutational signatures, other signatures are found only in some segments of the gastrointestinal tract. SBS2 and SBS13, due to activity of APOBEC cytidine deaminases, are common in small-intestine epithelial cells but rarely found in the oesophagus, stomach or large intestine, probably because of high APOBEC1 activity in small-intestine epithelial cells[7]. Similarly, SBS88, due to exposure to colibactin, a mutagenic product of a strain of *Escherichia coli* present in the microbiome, is often found in large-intestine epithelial cells but rarely elsewhere in the gastrointestinal tract[6,30]. As shown here, SBS17a and SBS17b are occasionally found in normal gastric epithelium but have not been reported elsewhere in normal cells of the gastrointestinal tract. This degree of similarity in mutational processes between segments of gastrointestinal tract is presumably a testament to the effectiveness of the various protective mechanisms operative between luminal contents and epithelial stem cells.

In contrast to the modest differences in SNV and indel mutation patterns (excepting metaplasia), the gastric epithelium carries recurrently generated trisomies of chromosomes 7, 13, 18 and 20 in a subset of individuals, a highly distinctive pattern that is not found in other sectors of the gastrointestinal tract or in cell types outside the gastrointestinal tract. The pattern of several generated trisomies of just a subset of different chromosomes in a subset of individuals raises the possibility of a microenvironment that increases the chromosome duplication rate or selects stem cells with these trisomic chromosomes to colonize glands and clonally expand. Nevertheless, glands with trisomies were not enriched in individuals with gastric cancer and did not carry particular driver mutations but can co-occur with metaplasia. Although a link with inflammation is probable, the precise nature of the instigating stimulus is unclear.

The landscape of cell clones with driver mutations in known cancer genes also differs markedly between the four segments of the gastrointestinal tract. In the oesophagus, about 60% of the normal squamous epithelium in 60-year-old individuals is occupied by cell clones with driver mutations[5,29]. In small- and large-intestine crypts, this proportion is much lower, about 1% (refs. 6,7), whereas 5% of the gastric glandular epithelium is occupied by clones with driver mutations. These differences may, at least in part, reflect the epithelial architecture, with the continuous stratified squamous epithelial sheet of the oesophagus allowing lateral spread of clones, whereas the crypt structure of the small intestine and large intestine hinders clones, with drivers spreading beyond the confines of the individual crypt. In the stomach, the branching structure of pits into tubules, and perhaps iterative damage and repair, may allow wider colonization of the epithelial lining than in the intestine. The sets of frequently mutated genes also differ between the different epithelia, with *NOTCH1*, *NOTCH2* and *TP53*, encoding proteins involved in wound healing, cell proliferation and DNA damage responses, dominating in the oesophagus, and genes encoding subunits of chromatin remodelling complexes, regulators of histone methylation and cell adhesion proteins dominating in the stomach, a repertoire still different from, but nevertheless more reminiscent of, mutated genes under selection in normal bladder epithelium[12].

The study explores changes in somatic mutagenesis associated with very early stages of neoplastic change in the stomach. Gastric glands

with IM, which are often associated with chronic inflammation and local clonal expansions, show increases in total mutation burdens due to elevated SBS1 (methylcytosine deamination), SBS18 (reactive oxygen species), ID1 and ID2 (replication strand slippage) and intrachromosomal structural variant mutation rates. These changes in mutagenesis could reflect an increase in cell division rates in metaplastic glands, be the consequence of other factors intrinsic to such cells or be due to microenvironmental influences, such as adjacent chronic inflammation. Large-intestine epithelial cells in areas affected by the inflammatory bowel diseases Crohn's and ulcerative colitis also show elevated mutation burdens with increased proportions of SBS18 mutations[31]. However, the close proximity of gastric metaplastic glands with elevated burdens to non-metaplastic glands with expected burdens and the presence of gastric glands in areas of chronic inflammation without elevated mutation burdens leaves the relationship between inflammation and mutation rate in the stomach uncertain. The elevated mutation burdens in metaplastic glands conceivably contribute to the increased cancer risk associated with IM.

Severe chronic inflammation was significantly associated with elevated numbers of driver mutations in gastric glands and overall proportions of mutant epithelium in this study, highlighting a role for chronic inflammation in moulding the preneoplastic selection landscape, as also identified in inflammatory bowel disease[31]. Beyond a role for inflammation, the large variation in this proportion across donors may indicate between-donor variation in selective pressures. Larger studies may be powered to find exposures or other agents that further impose the selection of specific clones, as has been found for smoking in the oesophagus[5], and promote the transformation of normal cells to overt tumours by means of metaplasia and dysplasia.

Gastric epithelial cells, therefore, exhibit a landscape of somatic mutations with some similarities to and many differences from those of other gastrointestinal epithelia. The differences probably reflect differences in intrinsic cell biology, tissue architecture, gut contents, processes of early neoplastic change and influences unknown at present, collectively contributing to shaping somatic mutational landscapes.

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

## Methods

### Ethics statement and sample collection
Snap-frozen gastric biopsy samples were obtained from three sources:
1. Multisite sampling performed on gastrectomy specimens removed as part of either gastric cancer treatment or bariatric surgery. Written informed consent for participation in research and publication of data was obtained from all donors in accordance with the Declaration of Helsinki and protocols approved by the relevant research ethics committees (RECs): (1) source country approval by the IRB of the University of Hong Kong/Hospital Authority of Hong Kong West Cluster, REC approval reference number UW14-257; (2) UK NHS REC approval from the West Midlands-Coventry and Warwickshire REC, approval number 17/WM/0295, UK Integrated Research Application System project ID 228343.
2. Multiregion gastric biopsies from transplant organ donors with informed consent for participation in research and publication of data obtained from the donor's family as part of the Cambridge Biorepository for Translational Medicine programme (UK NHS REC approval reference number 15/EE/0152; approved by NRES Committee East of England−Cambridge South).
3. Gastric samples obtained at autopsy from AmsBio (commercial supplier). UK NHS REC approving the use of these samples: London-Surrey Research Ethics Committee, REC approval reference number 17/LO/1801.

Further metadata for donors can be found in Supplementary Table 1, and metadata for all samples can be found in Supplementary Table 2 (WGS) and Supplementary Table 3 (targeted panel sequencing). No statistics were used to predetermine sample size, nor did we use any blinding or randomization.

### Laser capture microdissection and low-input DNA sequencing
Gastric tissue biopsies were embedded, sectioned and stained for microdissection as described in detail previously[2]. DNA libraries were constructed from microdissections using enzymatic fragmentation and subsequently submitted for WGS or targeted panel sequencing on the Illumina HiSeq X Ten or NovaSeq platform. Average sequencing coverage can be found in Supplementary Table 2 (WGS) and Supplementary Table 3 (targeted panel sequencing).

We used a custom Agilent SureSelect bait set to capture the exonic regions of 321 cancer-associated genes (listed in Supplementary Table 7).

### DNA sequence processing, mutation calling and filtering
DNA sequences were aligned to the GRCh38 reference genome by the Burrows−Wheeler algorithm[32]. SNVs and indels were called against the reference genome using CaVEMan[33] and Pindel[34], respectively. CNVs and structural variants were called using GRIDSS[35] and ASCAT[36] and are listed in Supplementary Table 4 (CNVs, both GRIDSS and ASCAT) and Supplementary Table 5 (intrachromosomal structural variants, GRIDSS only).

Beyond the standard postprocessing filters of CaVEMan, we removed variants affected mapping artefacts associated with the Burrows−Wheeler algorithm by setting the median alignment score of reads supporting a mutation as greater than or equal to 140 (alignment score median (ASMD) ≥ 140) and requiring that fewer than half of the reads were clipped (clipping median (CLPM) = 0).

We force-called the SNVs and indels that were called in any sample across all samples from a given donor, using a cut-off for read mapping quality (30) and base quality (25), before applying the Sequoia pipeline[37] for mutation filtering and phylogeny reconstruction.

As part of the mutation filtering, germline variants were removed using a one-sided binomial exact test on the number of variant reads and depth present across largely diploid samples, as previously described[3,8]. Resulting $P$ values were corrected for multiple testing with the Benjamini−Hochberg method, and a cut-off was set at $q < 10^{-5}$. To filter out recurrent SNV and indel artefacts, we fitted a beta-binomial distribution to the number of reads supporting variants and the total depth across samples from the same individual. For every indel or SNV, we estimate the maximum-likelihood overdispersion parameter ($\rho$) (ranging the value of $\log_{10}(\rho)$ from −6 to −0.05 in steps of 0.05). As artefactual variants appear in random reads across samples, they are best captured by low overdispersion, whereas true somatic SNVs and indels will manifest with high VAFs in some but are completely absent from other samples and are therefore highly overdispersed. To distinguish artefacts from true variants, we used $\rho = 0.1$ as a threshold for SNVs and $\rho = 0.15$ for indels, below which variants were considered to be artefacts. This filtering approach is an adaptation of the Shearwater variant caller[38].

We used a truncated binomial mixture model to model each whole-genome sample as a mixture of clones, determine the underlying VAF peaks and the corresponding clonality of the sample as previously described[3,37]. The truncated distribution is necessary to reflect the minimum number of reads that support a variant ($n = 4$) that is imposed by variant callers such as CaVEMan.

Phylogenetic trees were reconstructed using Sequoia[37], which uses a maximum parsimony framework as implemented in MPBoot[39], with default settings. Mutation mapping to branches was done using the treemut R package.

### Mutation rate analysis
To correct for the confounding of sequencing depth and detected number of mutations, we corrected the observed mutation burden by dividing over the estimated sensitivity. The sensitivity was estimated as the probability of observing a variant in at least four reads given the underlying coverage distribution per sample and the observed variant allele frequency peak per sample. The mean estimated sensitivity was 0.95 and the median 0.97. Raw and adjusted mutation burden estimates, for both indels and SNVs, are listed in Supplementary Table 2.

To estimate the mutation rate in normal gastric epithelium, we used a linear mixed-effects model, with age as a fixed effect and the donor as a random effect, on mutation burden estimates from gastric glands of non-cancer donors:
1. Burden - (distributed as) Age + (1|Donor)
   We assessed the effects of chronic inflammation (CI, coded as absent or mild versus moderate or severe) and IM (absent or present) by using these alternative models:
2. Burden - Age + CI + (1|Donor)
3. Burden - Age + IM + (1|Donor)
4. Burden - Age + IM + CI + (1|Donor)
   Models 2 and 3 significantly outperform model 1 ($P < 2.2 \times 10^{-16}$), and although model 4 outperforms model 2 ($P < 2.2 \times 10^{-16}$), it does not significantly outperform model 3 ($P = 0.11$). Therefore, presence or absence of IM and age predict mutation burden the best.
   To test the effect of gastric site on the mutation rate, we included site-specific age relations in the mixed-effects model:
5. Burden - Age:Site + IM + (1|Donor).

This model did not significantly outperform model 2 ($P = 0.4547$).

### Telomere length analysis
The average telomere length for WGS samples was estimated using the telomerecat algorithm[40], which uses the prevalence of the TTAGGG telomeric repeat in reads. Samples sequenced on the NovaSeq platform were excluded from this analysis, as we previously observed discordant results[1], such as telomere lengths of 0 bp, from such samples.

Similarly to the mutation burden analysis, we used a linear mixed-effect model to assess the effects of age, IM and chronic inflammation on telomere length:
1. Telomere length - Age + (1|Donor)

2. Telomere length ~ Age + CI + (1|Donor)
3. Telomere length ~ Age + IM + (1|Donor)
4. Telomere length ~ Age + IM + CI + (1|Donor)

Models 2 significantly outperforms model 1 ($P = 0.0005$), and model 4 does not significantly outperform model 2 ($P = 0.14$). Including IM annotation does not improve the model fit. Therefore, chronic inflammation and age predict telomere length the best.

## Mutational signature analysis

To identify possibly undiscovered mutational signatures in human placenta, we ran the hierarchical Dirichlet process (HDP) package (https://github.com/nicolaroberts/hdp) on the 96 trinucleotide counts of all microdissected samples, divided into separate branches of the phylogenetic trees. To avoid overfitting, branches with fewer than 50 mutations were not included in the signature extraction. HDP was run with the different donors as the hierarchy, with 20 independent chains, 40,000 iterations and a burn-in of 20,000.

The resulting signatures from HDP were further deconvolved into linear combinations of known COSMIC reference signatures (v.3.4) using an expectation-maximization mixture model. The deconvolution is accepted if the resulting linear combination of reference signatures has a cosine similarity to the original, extracted signature that exceeds 0.90. This resulted in the deconvolution of the HDP signatures into reference signatures SBS1, SBS2, SBS3, SBS5, SBS13, SBS17a, SBS17b, SBS18, SBS28, SBS40a and SBS40c. These signatures were then fitted to all observed SNV counts from individual branches using SigFit[41]. Signature exposures per sample can be found in Supplementary Table 2.

Of note, SBS5 and SBS40 have relatively flat and featureless mutation profiles, can be difficult to separate from each other and are therefore combined in analyses, as in previous reports[1,42].

The fold increase of specific mutational signatures in metaplastic glands compared to non-metaplastic glands was estimated by

1. Calculating the observed number of mutations incurred by each signature by multiplying the sensitivity-corrected mutation burden with the estimated signature exposures per sample.
2. Calculating the expected number of mutations incurred by each signature by multiplying the expected mutation burden, given the age of the donor and the average mutational signature distribution of all non-metaplastic glands of that donor. The latter accounts for any donor-specific differences in mutational signatures that may be present.
3. Dividing the observed over the expected mutation numbers per signature.

The procedure for indel signature extraction was performed identically to that listed above for SNVs. The five resulting HDP indel signatures were deconvolved into COSMIC reference signatures ID1, ID2, ID5, ID6, ID9, ID12 and ID14. HDP indel signature 5, a noisy signature typified by large deletions, was not decomposed further as no combination of reference signatures yielded a high enough cosine similarity to the extracted signature. As SigFit[41] is not compatible with indel signatures, the exposure to HDP signatures was converted to the exposure to reference signatures using the estimated signature proportions from the deconvolution. Signature exposures per sample can be found in Supplementary Table 2.

## Selection analysis and driver annotation

We used the dNdScv[27] R package to identify genes under positive selection. For genes in the targeted panel, we combined the data from both the WGS data and the targeted sequencing data. As the difference in coverage/clonality between the WGS data and the targeted sequencing data affects both non-synonymous and synonymous mutations, this mixed data can be safely fed into dNdScv, as applied previously[12]. To avoid overestimating the occurrence of mutations due to sampling

the same clone in several microdissections, we only count a specific mutation once per donor. Genes with a $q$-value below 0.1 were considered to be under selection, which amounted to *ARID1A*, *ARID1B*, *ARID2*, *CTNNB1*, *EEF1A1*, *LIPF* and *KDM6A*.

In addition, we used the WGS data to look for signs of selection outside of the genes included in the panel. This did not yield any further genes under selection.

To identify mutations in genes that are associated with cancer but did not appear in the positive selection analysis, we reviewed all mutations for canonical cancer driver mutations and annotated likely candidates. In brief, this involved annotating hotspot mutations in oncogenes and inactivating mutations (nonsense, missense and frameshift indels) in tumour suppressor genes through interrogation of the COSMIC database. Annotated driver mutations are listed in Supplementary Table 6.

## Estimate of proportion of epithelium with driver mutation

To estimate the proportion of gastric epithelium that harboured a driver mutation, we relied on the measurements of area of microdissections and the estimated cell fraction that harbours a driver mutation in a sample. This cell fraction is straightforwardly obtained by multiplying the VAF of a mutation by the local ploidy (1 for sex chromosomes in male donors, 2 otherwise). Multiplying mutant cell fractions with the area of epithelium sampled gives an estimate of the area of mutant epithelium. By summing up the total area of mutant epithelium and dividing by the total epithelial area sampled, we arrive at an estimate of the fraction of gastric epithelium that has been colonized by a clone with a driver mutation. For this analysis, we used both disrupting (missense, nonsense, frameshift and splicing) mutations in genes identified as under selection by dNdScv and manually annotated driver mutations (see the previous section).

Note that this approach assumes that the entire microdissected area consists of gastric epithelium. Any contaminant cell type will lower the VAF of mutations in gastric clones and therefore reduce the estimated mutant sizes. Therefore, the estimated proportion of epithelium can be slightly underestimated.

The role of age, chronic inflammation and IM on the proportion of gastric epithelium was evaluated using a set of linear models:

1. Mutant_proportion ~ Age
2. Mutant_proportion ~ Age + CI_grade
3. Mutant_proportion ~ Age + IM_proportion

These models were compared against each other using a two-sided ANOVA test. Model 2 was a significantly better fit than model 1 ($P = 0.002$). Model 3 was not a significant improvement over model 1 ($P = 0.86$). The effect of age was significant in all models, and severe chronic inflammation was significant in model 2 ($P = 0.001$).

## CNV timing

Assuming a constant mutation rate, the acquisition of large copy-number duplications, such as trisomies or events causing copy-number neutral loss of heterozygosity, can be timed by comparing the proportion of SNVs acquired before and after the duplication. These proportions can be estimated by clustering SNVs on the basis of their VAF. As previously, we used a binomial mixture model, using the counts of variant-supporting and total reads, to estimate the fraction of duplicated and non-duplicated mutations. Mutation clusters were assigned to be either duplicated or non-duplicated on the basis of the expected VAF from the CNV. For example, for a trisomy, the two VAF clusters would correspond to two different copy-number states: 0.66 (duplicated, mutations on two out of three copies) and 0.33 (non-duplicated, mutations on one out of three copies).

From the duplicated ($P_D$) and non-duplicated ($P_{ND}$) proportions, the total copy number ($CN_{total}$ and the duplicated copy number ($CN_{dup}$), the timing of the CNV ($T$) can then be estimated as follows:

$$T = \frac{CN_{total}}{CN_{dup} + \frac{P_{ND}}{P_D}}$$

The value of the CNV timing will be between 0 and 1, which—in the case of phylogenetic trees used here—corresponds to the beginning and end of the branch on which the CNV was acquired. To obtain a confidence interval around the single timepoint estimate, we used an exact Poisson test on the rounded duplicated and non-duplicated mutation counts.

### Identifying trisomy 20 in panel sequencing data in PD40293

Calling copy-number variation de novo from targeted panel sequencing is challenging due to the sparse sampling of the genome and uneven capture by baits, resulting in a very uneven coverage landscape. However, we used the paired WGS data to effectively assign SNPs on chromosome 20 to either parental haplotype, as these SNPs would have differential VAFs in the whole-genome sequenced samples with trisomy 20 (seven out of 12 samples) in PD40293, the donor with the most widespread recurrence of trisomy 20.

To identify the panel-sequenced microdissections with likely trisomy 20, we quantified the count data (reads supporting SNP and total reads) for phased SNP sites on chromosome 20. Using a likelihood ratio test, we quantified whether it was more likely that the two groups of phased SNPs were drawn from one binomial distribution with underlying probability of the mean total VAF across both haplotypes (diploidy) or two binomial distributions with a different probability underlying each of the two haplotypes (aneuploidy). To reduce noise, only SNP sites with more than five reads were used in this analysis. Resulting $P$ values were corrected for multiple testing using the Benjamini–Hochberg method. Besides significant differences, samples also needed to show a high enough difference between mean VAFs of the two haplotypes to be considered showing evidence of trisomy 20 (0.1).

### Reporting summary

Further information on research design is available in the Nature Portfolio Reporting Summary linked to this article.

## Data availability

DNA sequencing data have been deposited in the European Genome-Phenome Archive (EGA) with accession codes EGAD00001015351 (WGS) and EGAD00001015352 (targeted panel sequencing). Processed data are available in the Supplementary Tables or on GitHub (https://github.com/TimCoorens/Stomach; filtered variant calls and phylogenies). Reference genome GRCh38 is widely available (including at https://www.ncbi.nlm.nih.gov/datasets/genome/GCF_000001405.26/).

## Code availability

Custom R scripts for data analysis, filtering and visualization can be found at https://github.com/TimCoorens/Stomach.

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

**Acknowledgements** This research is funded by the Wellcome Trust, the Kadoorie Charitable Foundation and the Centre for Oncology and Immunology under the Health@InnoHK Initiative funded by the Innovation and Technology Commission, The Government of Hong Kong SAR, China. T.H.H.C. is the recipient of an EMBO long-term fellowship (grant no. ALTF 172-2022). The funders had no role in study design, data collection and analysis, decision to publish or preparation of the manuscript. We thank the staff of the Wellcome Sanger Institute Sample Logistics, Genotyping, Pulldown, Sequencing and Informatics facilities for their support with sample management and laboratory work. We thank K. Ardlie, S. Behjati, G. Getz and A. Lawson for discussions and critical review of the manuscript. We are grateful to the deceased transplant donors and their families for the gift of tissue donation facilitated by the Cambridge Biorepository for Translational Medicine. We thank the patients for participation in this study and the clinicians in the Hong Kong Hospital Authority for clinical care.

**Author contributions** G.C., S.Y.L. and M.R.S. designed the study. M.R.S and S.Y.L obtained funding. T.H.H.C. performed the analyses with help or input from G.C., H.J. and Y.W. K.M., K.S.-P., H.H.N.Y., S.Y.L. and S.Y.K.L. coordinated patient data and tissue samples. G.C. performed the microdissections with support from L.M. and Y.H. S.Y.L. and S.T.Y. analysed pathology in relation to genomic data. M.R.S. and S.Y.L. oversaw the study, with input from I.M. and P.J.C. T.H.H.C. and M.R.S. wrote the manuscript with input from all other authors.

**Competing interests** I.M., P.J.C and M.R.S. are cofounders, stockholders and consultants for Quotient Therapeutics Ltd. S.Y.L. has received research sponsorships from Pfizer, Merck and Servier. The other authors declare no competing interests.

**Additional information**
**Correspondence and requests for materials** should be addressed to Tim H. H. Coorens, Suet Yi Leung or Michael R. Stratton.

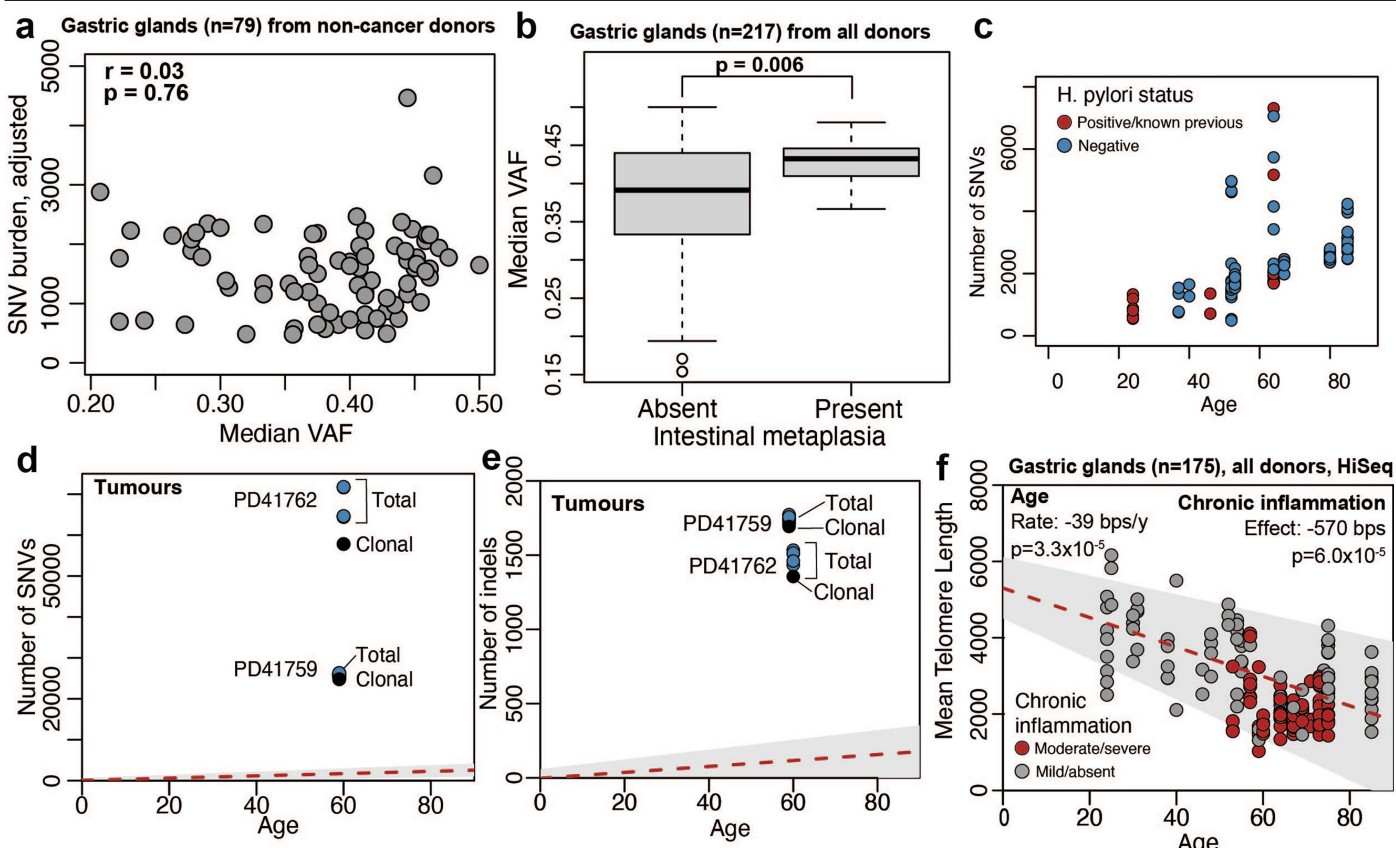

**Extended Data Fig. 1 | Clonality, mutation burden extended and telomere lengths. a**, SNV burden, adjusted for sensitivity using sequencing depth and clonality, against the median VAF in microdissections from non-cancer donors, showing no residual correlation between clonality and SNV burden. P-value obtained from Pearson correlation test. **b**, median VAF in microdissections from all donors, annotation for the presence of intestinal metaplasia, showing that metasplastic glands have significantly higher median VAFs. P-value obtained through a two-sided Wilcoxon rank sum test. The central line, box and whiskers represent the median, IQR from first to third quartiles, and 1.5 × IQR, respectively. **c**, SNV mutation burden in glands with confirmed *H. pylori* status. **d-e**, Detected burden of (**d**) SNVs and (**e**) indels in microdissections from

gastric cancers of PD41759 and PD41762, with the clonal SNV and indel burden, present in all cancer samples, denoted by the black dot. The red dashed line indicates the estimated age and SNV and indel mutation burden relation estimated from a mixed effects model in gastric glands from non-cancer donors (**Fig. d**), with the grey area indicating a confidence interval. (**f**) Estimated mean telomere length versus age for gastric glands. The red dashed line indicates the maximum likelihood age and telomere length relation estimated from a mixed effects model, with the grey shaded area indicating the 95% confidence interval. Glands with moderate or severe chronic inflammation show significantly shorter telomeres than those without. P-values obtained through a two-sided ANOVA test.

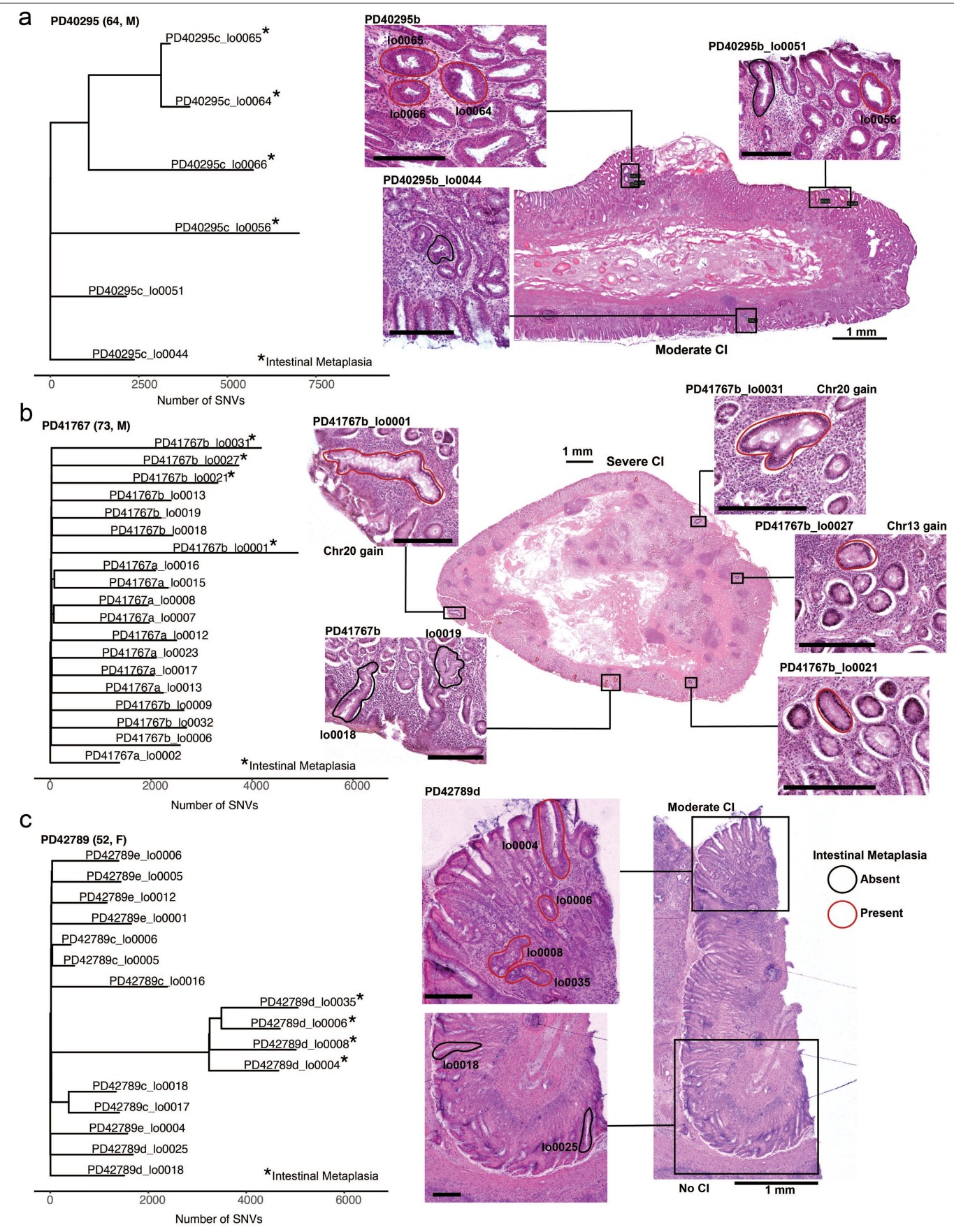

**Extended Data Fig. 2 | Phylogenies and histology of metaplastic glands.** Phylogenetic trees of three donors, PD40295 (**a**), PD416767 (**b**) and PD42789 (**c**), with metaplastic glands (indicated by the asterisk), along images of histology, with laser capture microdissections marked in black and pathological gradings. All biopsies are from the gastric antrum. CI=chronic inflammation, IM=intestinal metaplasia.

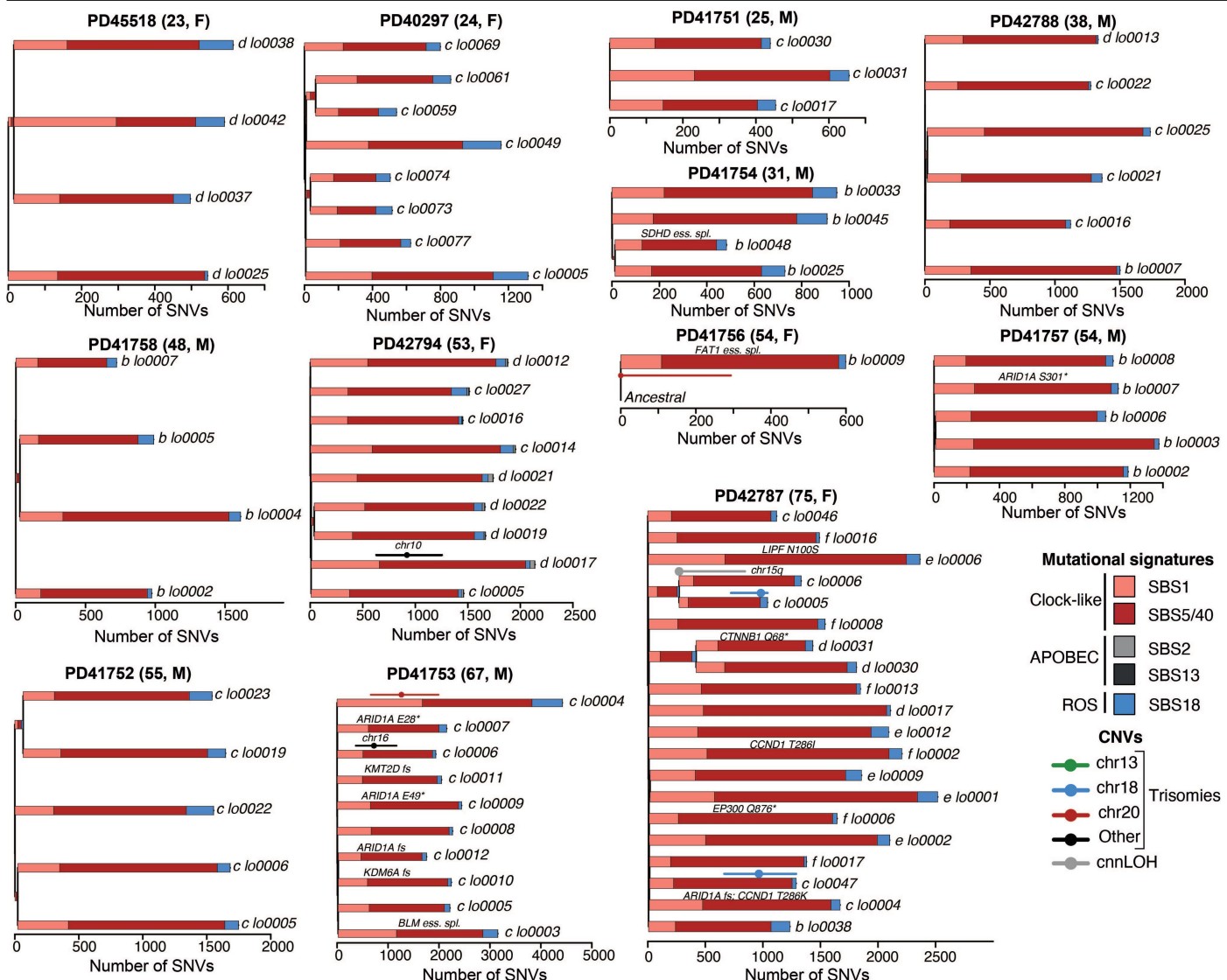

**Extended Data Fig. 3 | Mutational signatures in non-cancer donors.** Phylogenetic trees of gastric glands from non-cancer donors, branch lengths indicate the number of SNVs, and barplot per branch indicate the mutational signature proportion. Asterisks denote metaplastic glands.

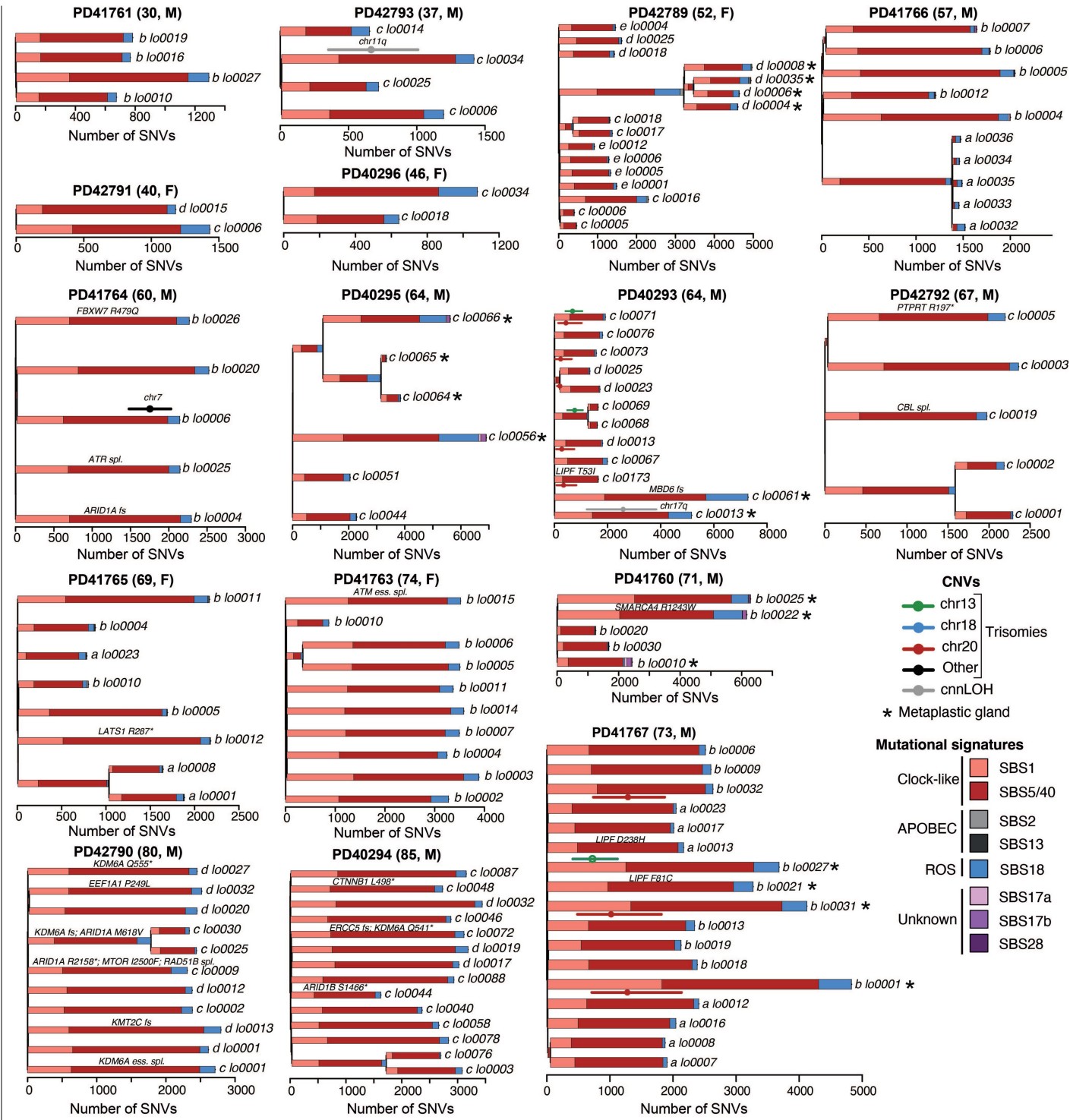

**Extended Data Fig. 4 | Mutational signatures in cancer donors.** Phylogenetic trees of gastric glands from cancer donors, branch lengths indicate the number of SNVs, and barplot per branch indicate the mutational signature proportion. Asterisks denote metaplastic glands. Note that the phylogenies for the four remaining cancer donors are shown in Fig. 2a–d.

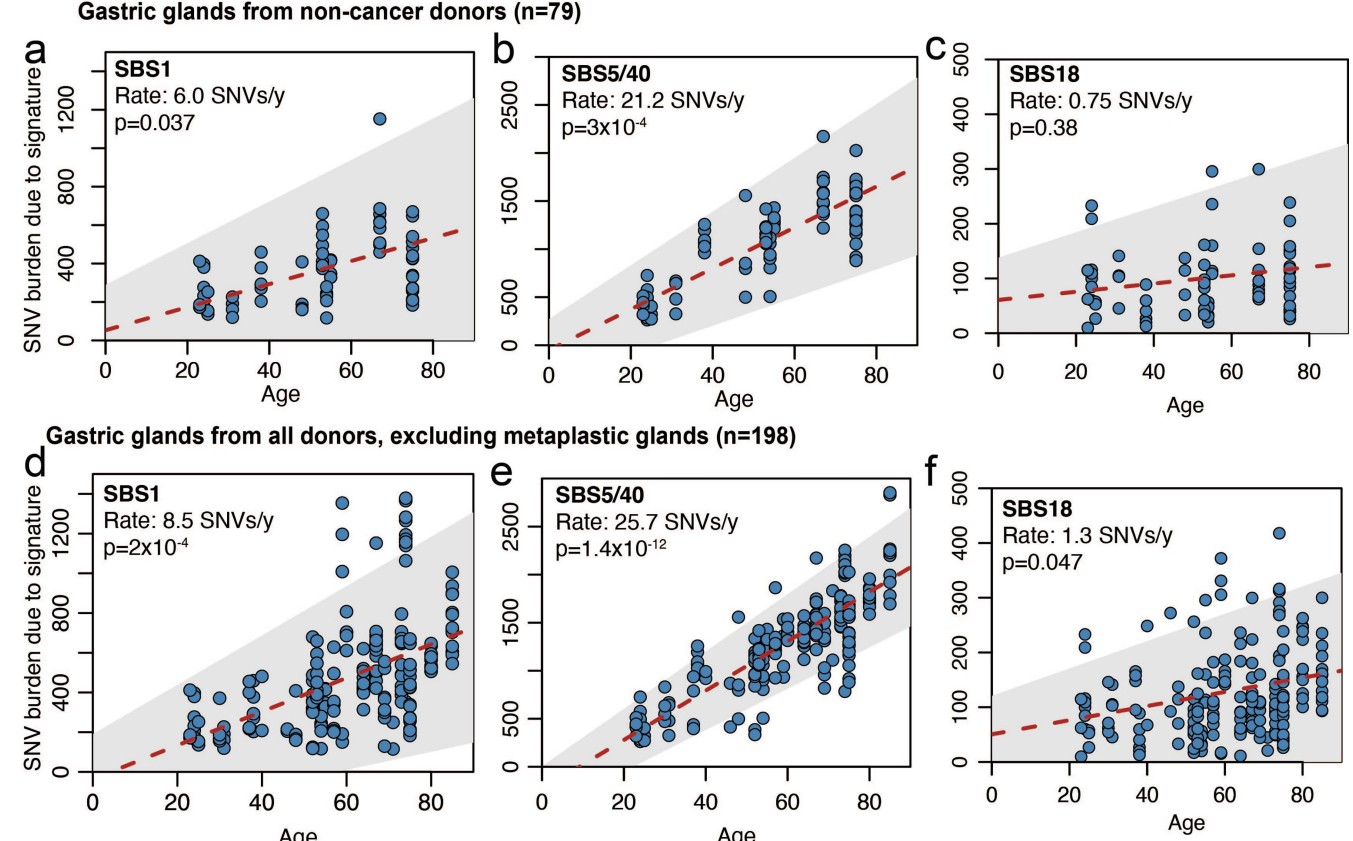

**Extended Data Fig. 5 | Accumulation of signature-specific burdens.** Burden of SNVs due to three signatures ubiquitous in gastric epithelium (**a**) SBS1, (**b**) SBS4/50 and (**c**) SBS18 versus age for gastric glands from non-cancer donors. Signature-specific burdens are also shown for gastric glands from all donors (excluding metaplastic glands) for (**d**) SBS1, (**e**) SBS5/40 and (**f**) SBS18. In all plots, the red dashed line indicates the maximum likelihood relation between age and SNV mutation burden due to a specific mutational signature obtained from a mixed effects model, with the grey box indicating a confidence interval. All P-values are obtained through a two-sided ANOVA test.

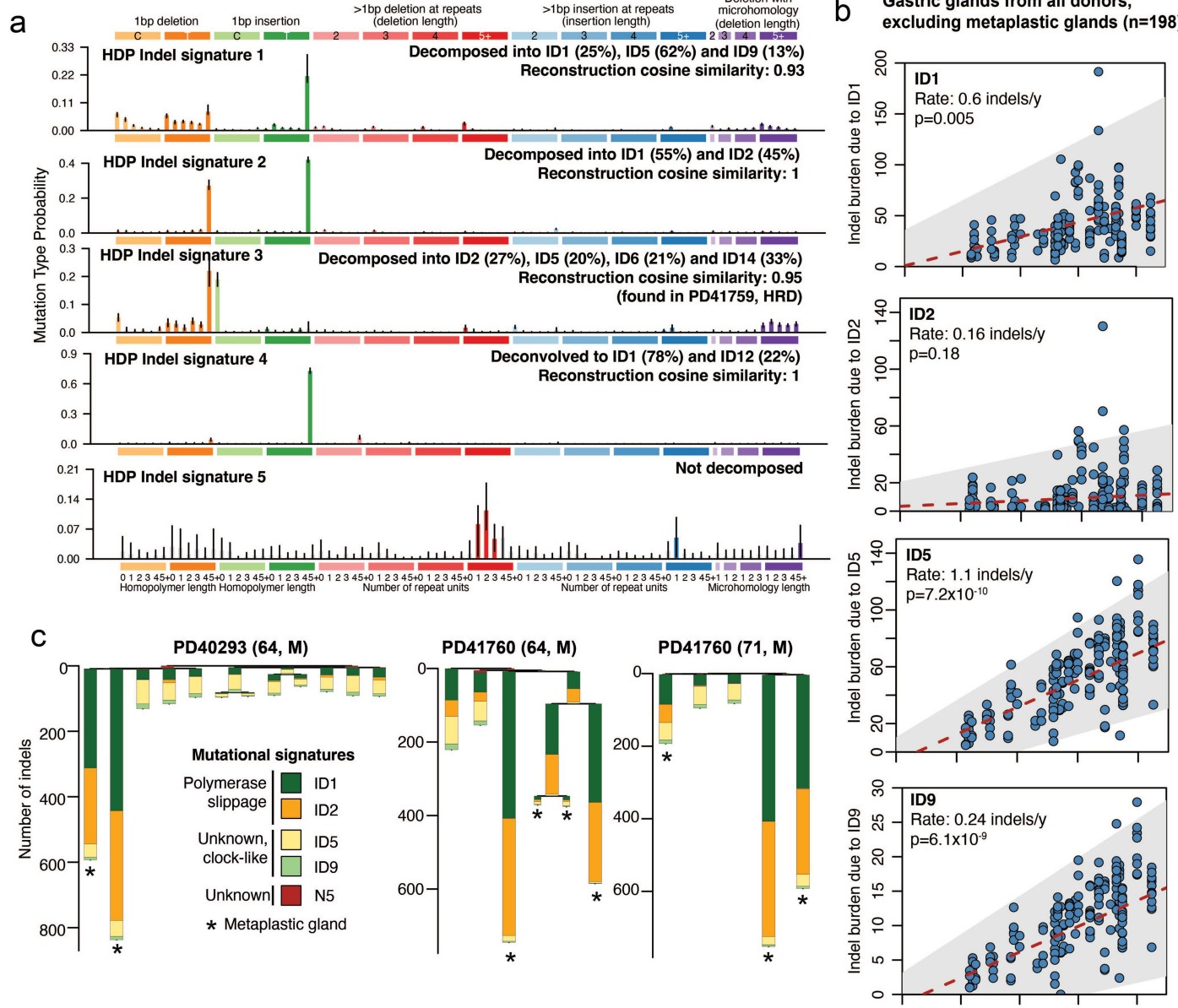

**Extended Data Fig. 6 | Mutational patterns of indels. a**, Indel signature extraction with HDP identified five signatures, which - with the exception of signature 5 - could be robustly deconvolved into a linear combination of reference indel signatures from COSMIC. Error bars denote 95% confidence interval around signature contribution estimate of the Bayesian posterior distribution as obtained through HDP. **b**, Burden of indels due to four signatures ubiquitous in gastric epithelium for gastric glands from all donors (excluding metaplastic glands). The red dashed line indicates the maximum likelihood relation between age and indel burden due to a specific signature obtained from a mixed effects model, with the grey box indicating a confidence interval. P-values are obtained through two-sided ANOVA tests. **c**, Phylogenetic trees of gastric glands from three donors, with indel burden as branch length, and coloured by signatures. Asterisks denote metaplastic glands.

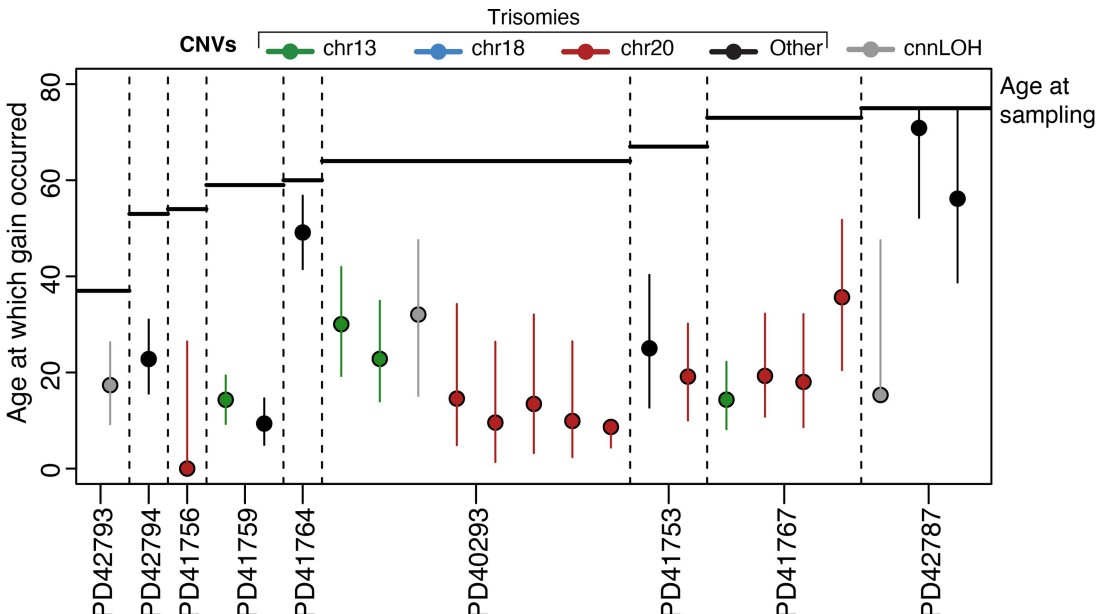

**Extended Data Fig. 7 | Estimate of donor age at which gain happened.** Estimated age of donor at which copy number gains occurred for each independent event, as calculated by dividing the estimated mutation burden at the gain by the total branch length and multiplied by the donor age at sampling, assuming a linear accumulation of mutations over age. Note that, if the most recent common ancestor of the clone/branch existed long before sampling, these estimates represent upper bounds. Confidence intervals represent Poisson confidence intervals calculated as part of the timing estimate and are based on the number of duplicated and non-duplicated SNVs in regions affected by CNVs (numbers listed in Supplementary Table 4). Black solid lines indicate donor age at sampling.

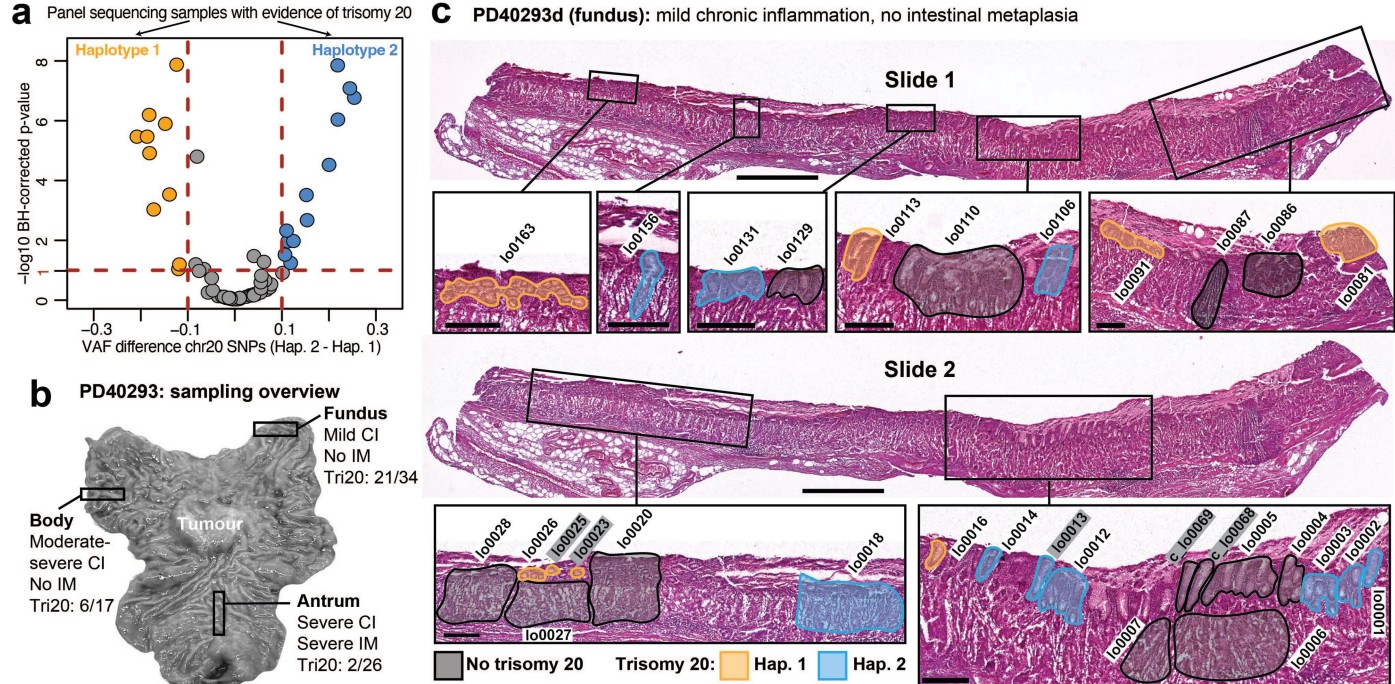

**a** Panel sequencing samples with evidence of trisomy 20

**c** PD40293d (fundus): mild chronic inflammation, no intestinal metaplasia

**b** PD40293: sampling overview

Fundus
Mild CI
No IM
Tri20: 21/34

Body
Moderate-
severe CI
No IM
Tri20: 6/17

Antrum
Severe CI
Severe IM
Tri20: 2/26

Tumour

No trisomy 20    Trisomy 20:    Hap. 1    Hap. 2

**Extended Data Fig. 8 | Widespread recurrent somatic trisomy 20 across one donor. a**, Scatter plot showing the difference in mean VAF of SNPs on both parental haplotypes of chromosome 20 per sample subjected to panel sequencing versus the Benjamini-Hochberg corrected p-value of a two-sided likelihood ratio test on read count data to test whether both haplotypes are more likely represented by two or one underlying binomial probabilities (**Methods**). Samples with an absolute VAF difference greater than 0.1 and a corrected p-value lower than 0.1 were deemed to show sufficient evidence of trisomy 20. **b**, High-level overview of the sampling of fundus, body and antrum in donor PD40293 with respect to other biopsies and the gastric tumour. CI=chronic inflammation, IM=intestinal metaplasia. **c**, Histology sections of the two histology slides with extensive microdissections of the fundus in PD40293 with annotated microdissected samples and presence of trisomy 20 based on (**a**). Grey box indicates samples has been whole-genome sequenced rather than subjected to panel sequencing. Scale bar in slide overview images are 1 mm, scale bars in zoomed-in images or 250 μm.

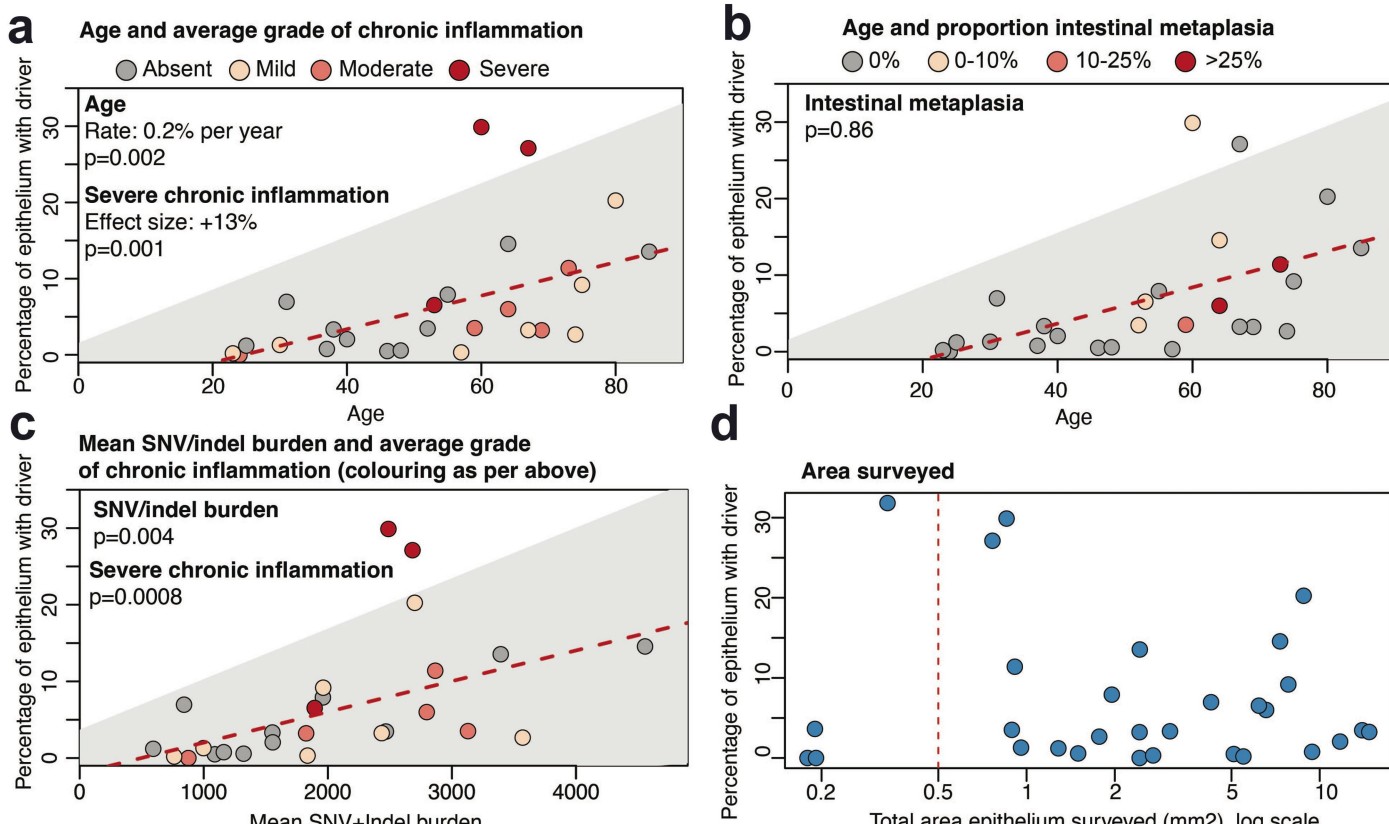

**Extended Data Fig. 9 | Driver proportion across age.** Percentage of epithelium with driver mutation versus age across donors with (**a**) chronic inflammation status and (**b**) proportion of intestinal metaplasia annotated. Only donors where more than 0.5mm² of epithelium was sampled were included in this analysis. The red dashed line indicates the maximum likelihood relation between age and percentage of gastric glands with drivers obtained from a mixed effects model, with the grey box indicating a confidence interval. P-values obtained through a two-sided ANOVA test. **c**, Similarly, scatterplot of percentage of epithelium with driver mutation versus small mutation burden (SNVs and indels) across donors, showing a consistent effect of severe chronic inflammation not explained by SNV/indel burden. The red dashed line indicates the maximum likelihood relation between mean mutation burden (indels and SNVs) and percentage of gastric glands with drivers obtained from a mixed effects model, with the grey box indicating a confidence interval. P-values obtained through a two-sided ANOVA test. **d**, Percentage of epithelium with drivers versus total area of gastric epithelium surveyed. Red dashed line indicates cut-off of 0.5 mm², below which donors were excluded from analysis in **a-c**.

# Reporting Summary

## Statistics

For all statistical analyses, confirm that the following items are present in the figure legend, table legend, main text, or Methods section.

| n/a | Confirmed | |
|---|---|---|
| ☐ | ☒ | The exact sample size (*n*) for each experimental group/condition, given as a discrete number and unit of measurement |
| ☐ | ☒ | A statement on whether measurements were taken from distinct samples or whether the same sample was measured repeatedly |
| ☐ | ☒ | The statistical test(s) used AND whether they are one- or two-sided<br>*Only common tests should be described solely by name; describe more complex techniques in the Methods section.* |
| ☐ | ☒ | A description of all covariates tested |
| ☐ | ☒ | A description of any assumptions or corrections, such as tests of normality and adjustment for multiple comparisons |
| ☐ | ☒ | A full description of the statistical parameters including central tendency (e.g. means) or other basic estimates (e.g. regression coefficient) AND variation (e.g. standard deviation) or associated estimates of uncertainty (e.g. confidence intervals) |
| ☐ | ☒ | For null hypothesis testing, the test statistic (e.g. $F$, $t$, $r$) with confidence intervals, effect sizes, degrees of freedom and $P$ value noted<br>*Give P values as exact values whenever suitable.* |
| ☐ | ☒ | For Bayesian analysis, information on the choice of priors and Markov chain Monte Carlo settings |
| ☒ | ☐ | For hierarchical and complex designs, identification of the appropriate level for tests and full reporting of outcomes |
| ☒ | ☐ | Estimates of effect sizes (e.g. Cohen's *d*, Pearson's *r*), indicating how they were calculated |

*Our web collection on statistics for biologists contains articles on many of the points above.*

## Software and code

Policy information about availability of computer code

| Data collection | No software was used for data collection. |
|---|---|
| Data analysis | - Alignment: BWA (https://github.com/lh3/bwa) (v0.7.17)<br>- SNV variant calling: CaVEMan (https://github.com/cancerit/CaVEMan) (v.1.14.0)<br>- Indel calling: Pindel (https://github.com/cancerit/cgpPindel) (v.3.9.0)<br>- CNV calling: ASCAT (https://github.com/cancerit/ascatNgs) (v.4.4.1)<br>- SV calling:  GRIDDS (https://github.com/PapenfussLab/gridss) (v.2.13.2)<br>- Mutational signature analysis: HDP (https://github.com/nicolaroberts/hdp) (v1)<br>- Mutational signature analysis: SigFit (https://github.com/kgori/sigfit) (v2.0.0)<br>- Phylogeny reconstruction: Sequoia (https://github.com/TimCoorens/Sequoia) (v1)<br>- Phylogeny reconstruction: MPBoot (https://github.com/diepthihoang/mpboot) (v1)<br>- Mutation mapping: treemut (https://github.com/NickWilliamsSanger/treemut) (v1)<br>- Telomere length estimation: TelomereCat (https://github.com/cancerit/telomerecat) (v4.0.1)<br><br>Custom R scripts for data analysis, filtering and visualization can be found at https://github.com/TimCoorens/Stomach |

For manuscripts utilizing custom algorithms or software that are central to the research but not yet described in published literature, software must be made available to editors and reviewers. We strongly encourage code deposition in a community repository (e.g. GitHub). See the Nature Portfolio guidelines for submitting code & software for further information.

# Data

Policy information about availability of data

All manuscripts must include a data availability statement. This statement should provide the following information, where applicable:

- Accession codes, unique identifiers, or web links for publicly available datasets
- A description of any restrictions on data availability
- For clinical datasets or third party data, please ensure that the statement adheres to our policy

> DNA sequencing data have been deposited in the European Genome-Phenome Archive (EGA) with accession codes EGAD00001015351 (whole-genome sequencing) and EGAD00001015352 (targeted panel sequencing). Processed data are available in the Supplementary Tables or on GitHub (https://github.com/TimCoorens/Stomach; filtered variant calls and phylogenies). Reference genome GRCh38 is widely available (including at https://www.ncbi.nlm.nih.gov/datasets/genome/GCF_000001405.26/).

# Research involving human participants, their data, or biological material

Policy information about studies with human participants or human data. See also policy information about sex, gender (identity/presentation), and sexual orientation and race, ethnicity and racism.

| | |
|---|---|
| Reporting on sex and gender | The study includes 12 donors of the female sex and 18 donors of the male sex. |
| Reporting on race, ethnicity, or other socially relevant groupings | The study includes 11 donors from Hong Kong, all of whom report South-East Asian ethnicity, and 3 donors from the UK and 16 from the US, all of whom are reported as "White" or "Caucasian" ethnicity. |
| Population characteristics | Data was obtained from 18 gastric cancer patients and 12 non-cancer donors. These donors span the US (n=16), UK (n=3) and Hong Kong (n=11), and are of various ages (between age 23 and 85) |
| Recruitment | 1. Multi-site sampling was performed on gastrectomy specimens removed either as part of gastric cancer treatment or bariatric surgery (Hong Kong University)<br>2. Multi-region gastric biopsies from transplant organ donors with informed consent for participation in research obtained from the donor's family as part of the Cambridge Biorepository for Translational Medicine program (UK)<br>3. Gastric samples obtained at autopsy from AmsBio (commercial supplier) (US) |
| Ethics oversight | Snap-frozen gastric biopsy samples were obtained from three sources:<br>1. Multi-site sampling was performed on gastrectomy specimens removed either as part of gastric cancer treatment or bariatric surgery. Written informed consent for participation in research was obtained from all donors in accordance with the Declaration of Helsinki and protocols approved by the relevant research ethics committees (RECs): (i) source country approval by the IRB of the University of Hong Kong/Hospital Authority of Hong Kong West Cluster, REC approval reference number UW14-257; (ii) UK NHS REC approval from the West Midlands-Coventry and Warwickshire REC, approval number 17/WM/0295, UK Integrated Research Application System (IRAS) project ID 228343.<br>2. Multi-region gastric biopsies from transplant organ donors with informed consent for participation in research obtained from the donor's family as part of the Cambridge Biorepository for Translational Medicine program (UK NHS REC approval reference number 15/EE/0152; approved by NRES Committee East of England – Cambridge South).<br>3. Gastric samples obtained at autopsy from AmsBio (commercial supplier). UK NHS REC approving the use of these samples: London-Surrey Research Ethics Committee, REC approval reference number 17/LO/1801. |

Note that full information on the approval of the study protocol must also be provided in the manuscript.

# Field-specific reporting

Please select the one below that is the best fit for your research. If you are not sure, read the appropriate sections before making your selection.

☒ Life sciences ☐ Behavioural & social sciences ☐ Ecological, evolutionary & environmental sciences

For a reference copy of the document with all sections, see nature.com/documents/nr-reporting-summary-flat.pdf

# Life sciences study design

All studies must disclose on these points even when the disclosure is negative.

| | |
|---|---|
| Sample size | The cohort consists of 30 individuals, 18 with gastric cancer and 12 with no gastric pathology, from Hong Kong, the United States or the United Kingdom. From these donors, 217 normal gastric glands and 21 neoplastic glands from the gastric cancers of two individuals were microdissected and individually whole genome sequenced to 23-fold median coverage. In addition, we subjected a further 829 microdissections comprising a total of 8,0007 gastric glands to targeted sequencing of known cancer genes. The numbers of individuals and glands sampled per individual are in line with previous efforts to map somatic mutation landscapes in colon (Lee-Six et al, 2019, Nature), endometrium (Moore et al., 2020, Nature) and placenta (Coorens et al., 2021, Nature) and so the sample sizes were deemed sufficient to achieve the aims of the study. |

| Data exclusions | Data with poor coverage after laser capture microdissection and whole-genome sequencing (< 10x) were excluded from analysis. |
|---|---|
| Replication | The LCM and sequencing pipeline, as well as the variant calling and data analysis pipelines, have been extensively replicated and validated across more than 10 studies. (Ellis et al., 2021, Nature Protocols; Moore et al., 2021, Nature; Brunner et al. ,2019, Nature; Lee-Six et al., 2019, Nature; Coorens et al. 2021, Nature; Lawson et al., 2020, Science; Moore et al., 2020, Nature; Robinson et al., 2021, Nature Genetics; Olafsson et al., 2020, Cell; Wang et al., 2022, Nature Genetics). |
| Randomization | Not applicable to this study - donor status as cancer patient or non-cancer donor was known a priori. Study describes the landscape of somatic mutations in gastric epithelium across individuals and hence is descriptive in nature, rather than testing a specific hypothesis in the population (which may have required randomization). |
| Blinding | Not applicable to this study - donor status as cancer patient or non-cancer donor was known a priori.Study describes the landscape of somatic mutations in gastric epithelium across individuals and hence is descriptive in nature, rather than testing a specific hypothesis in the population (which may have required blinding). |

# Reporting for specific materials, systems and methods

We require information from authors about some types of materials, experimental systems and methods used in many studies. Here, indicate whether each material, system or method listed is relevant to your study. If you are not sure if a list item applies to your research, read the appropriate section before selecting a response.

## Materials & experimental systems

| n/a | Involved in the study |
|---|---|
| ☒ | ☐ Antibodies |
| ☒ | ☐ Eukaryotic cell lines |
| ☒ | ☐ Palaeontology and archaeology |
| ☒ | ☐ Animals and other organisms |
| ☒ | ☐ Clinical data |
| ☒ | ☐ Dual use research of concern |
| ☒ | ☐ Plants |

## Methods

| n/a | Involved in the study |
|---|---|
| ☒ | ☐ ChIP-seq |
| ☒ | ☐ Flow cytometry |
| ☒ | ☐ MRI-based neuroimaging |

## Plants

| Seed stocks | *Report on the source of all seed stocks or other plant material used. If applicable, state the seed stock centre and catalogue number. If plant specimens were collected from the field, describe the collection location, date and sampling procedures.* |
|---|---|
| Novel plant genotypes | *Describe the methods by which all novel plant genotypes were produced. This includes those generated by transgenic approaches, gene editing, chemical/radiation-based mutagenesis and hybridization. For transgenic lines, describe the transformation method, the number of independent lines analyzed and the generation upon which experiments were performed. For gene-edited lines, describe the editor used, the endogenous sequence targeted for editing, the targeting guide RNA sequence (if applicable) and how the editor was applied.* |
| Authentication | *Describe any authentication procedures for each seed stock used or novel genotype generated. Describe any experiments used to assess the effect of a mutation and, where applicable, how potential secondary effects (e.g. second site T-DNA insertions, mosiacism, off-target gene editing) were examined.* |

