## [Peer Review File · Nature]

The somatic mutation landscape of normal gastric epithelium

Corresponding Author: Professor Michael Stratton

Version 0:

Reviewer comments:

Referee #1

(Remarks to the Author)

In this manuscript Coorens and co-workers describe their work analysing the somatic mutation type and burden of normal and (pre)malignant gastric glands. The methods follow an approach which is now well described focusing on ~30x WGS of microdissected individual clonal units in the luminal gut. The data are well-presented and I particularly like the way this study is framed against the background of previous studies from this group examining mutation accumulation in the oesophageal, small intestinal and large bowel epithelial lining. I feel this is an excellent piece of work.

I have two major points of critique and several minor points of concern for the authors' consideration. My two main points of critique focus on so-called hypermutant glands (1) and the polyclonal microbiopsies which have been analysed through targeted capture sequencing (2). I will address these in turn, with apologies for the (uncharacteristic) lengthy review.

Hypermutant glands: Here the authors make the case for a select group of glands which have gone through a period of accelerated mutation accumulation (as per line 125-127: 'it appears that each gland has independently increased its mutation rate during the lifetime of the individual in response to a local stomach environment in which a gastric cancer has developed'). These data overall are very persuasive – a population of glands clearly stands out, both in terms of mutation burden (Fig 2) and in terms of mutation signature distribution (Fig 3e). These glands are mainly found in the antrum (nb. although the antrum is regrettably under sampled in the non-cancer donors as per Fig 2, top panel) and are specifically associated with cancer donors. Importantly, it appears that nearly all hypermutant glands (except for one) derived from intestinal metaplasia. This latter point could have major implications for our understanding of gastric cancer progression, but I fear this message might be lost because of the semantics (see below).

Published evidence shows that intestinal metaplasia is a clonal precursor (PMID: 18242216) which progresses to dysplasia (PMID: 21223968). Although the authors suggest a temporal increase in mutation *rate* (my emphasis) to explain the increased mutation burden in these glands, no mechanistic evidence for this is provided. By contrast, the authors also state (line 200-202) that 'since SBS1 and SBS18 mutation rates have previously been associated with increased cell division rates and inflammation, it is plausible that the hypermutation in these glands reflects periods of increased stem cell proliferation'. This would instead suggest that the increased burden is not due to an increased mutation *rate*, but rather secondary to an increased cell turnover.

Evidence indicates that gastric glands undergo niche turnover through a process of neutral competition akin to colonic crypts, although details as to number of stem cells and clearance rates in gastric stem cell niches remain unclear. In *Helicobacter*-infected gastric epithelia often intense cytotoxic cell death is seen around the gastric isthmus stem cell zone and stem cell proliferation rates are clearly increased compared to uninfected gastric mucosa. Ultimately, this drives somatic selection and expansion of metaplastic lineages which have gone through a more recent bottleneck compared to any randomly-selected background gastric gland.

Overall, the most parsimonious explanation for the observed increased mutation burden in this 'hypermutant' category of glands is not increased mutation *rates*, but rather more likely decades-long accelerated turnover in a chronically inflamed environment coupled with clonal selection and expansion of metaplastic lineages. This interpretation of the genetic data in this study is akin to the two-step initiation-promotion process first described in mouse models and recently revived by the Swanton team (see PMID: 37020004 and PMID: 37059064) in the context of air pollution-linked EGFR mutation in chronically inflamed lung epithelia. In this model the first step is stochastic acquisition of an oncogenic mutation, often through endogenous clock-like mutational processes, and the second is exposure to an inflammatory stimulator, which drives lineage selection and expansion, in particular of those adapted to the inflammatory environment. The two steps together are required for tumour formation. These discoveries have important implications for preventative medicines targeting the first steps in tumour evolution. The authors' approach to interrogate mechanisms of early cancer initiation in

normal tissues is ideally suited to address this point and would make a highly impactful public health message, namely that gastric cancer initiation is due to chronic inflammation-associated accumulation of endogenous clock-like oncogenic processes in clonally selected lineages.

In this context I fear that 'hypermutant' glands suggests some burst-like mutational process which distracts from the underlying clonal dynamics. On a tangent to this, the authors also state that (line 124) 'Hypermutant glands were phylogenetically unrelated to each other beyond early development, except for one clonal expansion detected in PD42789 (Extended Data Fig. 1)'. It is unclear whether this statement rules out that clonal expansion in hypermutant lineages is limited to isolated examples (like PD42789), as this would require additional dense gland-by-gland sampling around an IM gland (low power images are not provided to assess this). I would suggest that the clonal expansion shown in Extended data 1c instead indicates that intestinal metaplasia can undergo sizeable clonal expansions (in line with PMID: 18242216). To summarise I would suggest the authors revise this description and preferably drop the potentially confusing term 'hypermutant' throughout. Instead, to drive mechanistic and clinical impact I would suggest the authors specifically label the metaplastic samples in Figure 1B, Figure 1C to F and Figure 2, and focus the discussion of their data on the accumulation of clock-like mutational processes in chronically inflamed epithelia (likely *Helicobacter*-associated) and eventual selection and expansion of intestinal metaplasia. Direct in situ demonstration of the clonal initiation of intestinal metaplasia would make a valuable addition to the manuscript.

Targeted capture data: Unlike the colonic mucosa which essentially consists of a monotonous repetition of clonal units, clearly spatially separated, regardless of the plane of sectioning, the gastric mucosa has a much more complex arrangement characterised by clonal units which extensively branch towards the base. Earlier studies using classic sporadic marker alleles have shown that these branches are clonally related and therefore aren't individual units (see for example PMID: 8062232). An important consequence of this is that the cellular progeny of a stem cell carrying a somatic variant likely streams into multiple individual glands/tubules.

Although it isn't spelled out directly, the wording in the manuscript suggests that each 'gland' (that is, a branch/tubule of a clonal unit) is counted as one clonal unit akin to a colonic crypt. Further complicating this issue is that individual glandular branches often mingle. In this way, each of the ± 800 microbiopsies contains on average 10 glands, but those 10 glands could all derive from a single stem cell niche or (more likely) from multiple neighbouring stem cell niches. These spatial considerations of course affect the interpretation of spatially clustered mutations, variant allele frequencies and modeling data.

My primary concern is that without some introduction to these caveats that the general reader may view these data to mean that 'glands' correspond with individual clonal units in the 'crypt' sense. I would suggest to (briefly) introduce these concepts and avoid the term 'gland' in this context since its meaning is ambiguous, in particular when analysing polyclonal input. For example, where the abstract states 'Surveying approximately 8,000 gastric glands by targeted sequencing' this could be misinterpreted as 8,000 individual clonal units. I would instead suggest 'Surveying approximately 829 polyclonal gastric microbiopsies by targeted sequencing' (cf. line 291 and Fig 5c). This mingling of glands and clonal units likely explains the spatial clustering of KDM6A and ARID1A variants (greater curve biopsy, Fig 5D top) and one of the beta-catenin variants (lesser curve, Fig 5D bottom). By extension, the maximum parsimony phylogeny shown in Fig 5D therefore isn't necessarily incorrect, but its meaning is different from the beautiful phylogenies in Fig 3. Again I feel this must be highlighted.

I suspect this spatial clustering effect also affects the calculation shown in Ext Data 7 (relation between age and percentage of gastric glands with drivers) and its interpretation ('in 60-year-old individuals approximately 5% of glands were colonised by clones with drivers'). The assumptions that underpin the model must be made clear. Finally, in a similar vein I suspect that since the dNdS calculations were derived by combining single gland WGS and targeted capture data from microbiopsies (Methods) that spatial mingling of clones and spurious recurrence of variants could also skew these data. This must be addressed. Overall I feel that these targeted capture data on polyclonal units must be presented with the proper caveats.

Further points:

- I suspect that the data in Fig 1B are from individual glands only, not patient-matched microbiopsies. I would suggest emphasising this point.

- Line 279/280: 'Glands with trisomies were not enriched in individuals with gastric cancer ($p=0.48$, Fisher's exact test) and were not significantly more likely to harbour driver mutations than glands without trisomies ($p=0.74$, Fisher's exact test).' The authors do not provide a dedicated discussion on driver mutation data from the 30x WGS gland set. This feels like a gap in the story and I would suggest this is addressed.

- Line 285: 'However, had a pathogen been implicated in trisomy acquisition, the infection may have predated the sampling by decades and would be unlikely to have persisted at readily detectable levels at the time of biopsy.' It is thought that most *Helicobacter*-positive patients are infected early in life (<10 yo) and that infection is maintained throughout life unless a patient is either treated or significant atrophy sets in which means *Helicobacter* can no longer thrive (*Helicobacter* autolyses at $pH>5$). Without direct serologic information, atrophy status or prior treatment status it is not possible to state that it is unlikely to have persisted and this statement must be toned down or qualified.

- Extended data figure 1 – the panels carry helpful histopathology annotations with regards the presence of chronic inflammation and grade of intestinal metaplasia. As stated before, I feel additional lower power overviews may be instructive to reveal the spatial density of sampling, also with regards to statements on the phylogenetic relatedness/clonal expansion of metaplastic clones. Second, international guidelines recommend that the extent of intestinal metaplasia across the stomach is graded, not the degree of involvement within a single gland.

Referee #2

(Remarks to the Author)

Tim Coorens and colleagues propose an impressive dataset of 238 whole-genome and 829 targeted sequencing of laser-microdissected gastric glands across 30 donors, including 18 cancer donors.

After similar studies on the human oesophagus and intestines, this is the first large study of the somatic evolution of the human stomach in the non-cancer and cancer contexts.

They focus on the evolution of 'normal' non-cancer glands in those contexts and contrast with 21 cancer microdissections.

First, most glands were monoclonal with VAF of the mutations close to 0.5 (>0.25). Akin to other organs, they show that the mutation rates increases linearly with age at a rate of 27.8 SNVs and 2 indels per year. Some glands, mostly in the antrum of 7 cancer donors, showed significant excess of SNVs relative to their age-related baseline, which they refer to as hypermutant glands. These hypermutators were unrelated phylogenetically, suggesting a common hypermutator environmental rather than genetic background. Indeed, most of these cases turned out to have chronic inflammation and/or metaplasia, while one case had no inflammation nor metaplasia. More generally, the SNV/indel burden in glands of cancer donors was significantly higher.

The mutations in normal glands followed patterns of known mutational processes, including clock-like (SBS1, SBS5/40), and ROS (SBS18). SBS1/5/50/18 were present in all glands, suggesting generic processes. Clocklike mutations accumulated linearly with age but not SBS18. The proportions of signatures related to replication, SBS1 and SBS18, increased in hypermutators, suggesting periods of higher proliferation leading to those. Indel proportion also increased with signatures linked to errors during replication. Whether this is due to metaplasia leading or in response to cancer is unclear. SBS17ab, which is rarely seen in normal tissues, but was seen in oesophagus and gastric cancers or Barrett's oesophagus, was present in a few normal glands of cancer donors, suggesting processes generating them in a normal context in the stomach. APOBEC signatures (SBS2/13) were also not seen in normal glands, unlike in gastric cancers.

Interestingly, unlike in other normal human tissues, CNVs were not rare, mostly smaller events at fragile sites, chromosome arm events leading to cnn-LOH, and trisomies of chr20 and chr13 in normal gastric glands, and could arise independently in glands from different branches of the phylogenies within the same donor. Thanks to gain-timing analyses, the authors show that gains of chr20, akin to its timing in gastric cancers, arose early in the patients lifetimes. They were not linked to cancer patients, a specific anatomical location, or the presence of driver mutations, but related to hypermutators.

Next, the authors profiled 321 known cancer genes with high-depth targeted sequencing in an extra 834 microdissections, each representing multiple glands. They found evidence of positive selection in 6 genes: ARID1A/B, CTNNB1, KDM6A, ERBB3, and LIPF, a gastric lipase. Except for LIPF, all were identified in gastric cancers. However, CTNNB1 had patterns reminiscent of tumour suppressor in normal glands rather than oncogene in cancer glands. Some of the driver patterns (e.g. one CTNNB1 mutation) showed converging evolution confined to mm space. Other likely drivers at known hotspot locations were found, e.g. BRAF, but not in TP53 or PIK3CA, two common drivers in gastric cancer. The number of drivers increased with age such that by age 60yo, 5% of the glands had a driver mutation.

Altogether, this is an important novel study for the field of human somatic evolution, describing the somatic evolution of the human stomach for the first time, across age groups and cancer vs. non-cancer donors. It is a purely genomics-based study, repeating analyses already performed in other organs and with no molecular phenotypic data. The originality stems from the exclusive dataset for the human stomach rather than the now well-established methods. I believe it is an important high-impact work that deserves publication.

It would be great if the authors could answer a few major comments and some minor comments for this work. Please see below.

Major Comments.

1. Line 685. Data availability. Please make sure to publish the data.
2. Line 681. Code availability. The github code should have a README with a clear description, without which it is unnecessarily challenging to reproduce the analyses.
3. As acknowledged by the authors, there is no antrum data in the non-cancer donors. But I do not follow the reasoning in the discussion. What is a "diseased antrum"? Do the authors mean that these hypermutant glands would be exclusively seen in the cancer context, or in a metaplasia/inflammation context? To me it could very well be that the antrum in the absence of cancer generally has regions with glands of high mutation load, potentially linked to metaplasia/inflammation. If this was the case, I do not think the current design would have allowed to see it. While the authors talk about an 'association' with metaplasia/inflammation, they do not show metaplasia/inflammation for the whole cohort, and do not show it is absent in non-cancer donors. I think the authors, if they could not add antrum data in the non-cancer donor group, should at least show/discuss the metaplasia/inflammation status in the other cancer donors and the non-cancer donors, which to my knowledge could very well be positive in a large proportion.

Other comments.

Line 86. "The mean variant allele fractions (VAFs) of somatic single nucleotide variants (SNVs) and small insertions and deletions (indels) generally exceeded 0.25 (Fig. 1b), indicating that gastric glands are predominantly monoclonal cell populations derived from recent single stem cell progenitors." Could the authors please detail the reasoning a bit more, i.e. on the rationale of the 0.25 threshold and its relation to monoclonality and the recency of the stem cell progenitor.

Fig 1cd. the n value is next to donors rather than glands, which is inaccurate.

Fig 1cdef. hypermutators outside 95% interval could already be highlighted (e.g. in red) to help with figure glancing.

Fig 1g. the label of the two groups under number should be something like "Glands, non-cancer donors", as otherwise the numbers seem to report to donors rather than glands again.

Ext Fig 2bc, Did you count only clonal mutations in the tumour microdissections - could it be that a large fraction of the excess is simply explained by later subclonal mutational events?

Fig3abc. We suggest removing border in the bars, such that the colour of smaller contributions still have a chance to appear (currently overshadowed by the black borders).

Regarding the relation of SBS18 with age. In the phylogenies where two glands originate from a "late" recent common ancestor, SBS18 mutations are mostly seen in the late branches, even in the relatively later branching in mutational time. This would contradict the hypothesis of early development processes. Could the authors comment on this. Related to this, could the authors also use the mutations at multiplicity 2 (vs. multiplicity 1) on trisomies to time SBS18 signatures in the early molecular-time periods?

Line 200. "Since SBS1 and SBS18 mutation rates have previously been associated with increased cell division rates and inflammation". Please provide a citation for this.

Fig 4b. This figure in a table format should also show information of cancer vs. non-cancer donors.

Line 273. "There was no significant effect of age on the burden of trisomies." This seems surprising from the table on Figure 4b, where clearly trisomies are enriched in older patients within sex-groups. Are CNVs more present in cancer donors? Could it be that the statistical test used is not powered to detect the enrichment clearly present in this table? Could it be most patients acquire CNV early in life, which are typically selected for later in life?

Line 282. "However, there was a modest enrichment of trisomies in hypermutated glands ($p=0.03$, Fisher's exact test), suggesting a role of factors causing chronic inflammation in their genesis." Why inflammation and not metaplasia instead?

Line 329. That there are 5% glands with drivers at 60yo: this is a useful number but it should come with a sense of the biological/technical variability. The confidence interval on this estimate is very large and the observed values for the few donors around 60yo range span an order of magnitude (from 1% to 10% ; extended data fig. 7). It does not seem like there is a sharp shift in the elderly like for the clonal diversity in haematopoiesis. Could the authors also comment on this variability and how it compares to other organs and segments of the gastrointestinal tract.

Line 355. "This degree of similarity in mutational processes between segments of gastrointestinal tract is presumably a testament to the effectiveness of the various protective mechanisms operative between luminal contents and epithelial stem cells". This implicitly says that we would expect differences in mutational process exposure between the segments, could the authors please elaborate a bit on those?

Line 392. "In the stomach, the gland structure, and perhaps iterative damage and repair, may allow wider colonisation of the epithelial lining than in the small and large intestine." Could the authors please elaborate on what aspects of the gland structure of the stomach would allow wider colonisation than that of the intestines.

Extended Data Figure 3/4. Please add all the gain timings to those if applicable and comment on them.

Telomere lengths. The authors did not but could look at telomere lengths across age, cancer vs. non-cancer, and how they relate to the CNVs, SNVs loads, presence of metaplasia etc.

Translocations, tandem duplications, inversions. Although the authors called them with GRIDSS, there is no mention of structural variants at all, just CNVs. Given that stomach glands are relatively instable and present CNVs, it is not clear if copy-neutral structural changes were not detected at all or were not reported by the authors. It would be interesting to see the load and types of SVs (even if the numbers are 0's), and contrast them with the cancers.

(Remarks on code availability)

The code looks ok but there is no README.

This is the second major comment in my comments to the authors:

2. Line 681. Code availability. The github code should have a README with a clear description, without which it is unnecessarily challenging to reproduce the analyses.

Referee #3

(Remarks to the Author)

This study by Coorens et al reports about somatic mutation landscapes in normal and in cancerous stomach epithelial cells, by sequencing (usually clonal) microdissections of gastric glands. It is a largely rigorous study, however see comments to be addressed below; similarly so there are some concerns with reporting. The novelty does not lie in the general approach, which mirrors their previous work on other organs (e.g. colon, uterus). Instead, the strength of the study are some rather striking findings on existence of hypermutant glands (which are unevenly distributed) and the not-uncommon copy number alterations observed in healthy cells and their very intriguing recurrence within an individual. While these two findings are based on a low number of observations (in terms of individuals) and thus in a sense anecdotal, in my opinion they do provide an important basis for future more comprehensive research of these phenomena. Other findings of some interest include the change in mutation spectra between normal and cancerous cells (this was known for APOBEC), as well as in the driver mutation spectra between normal and cancerous cells (the CTNNB1 association is indeed, as they write, intriguing). Studying premalignant events may yield insights relevant for cancer early detection and prevention. I have some concerns about the analyses and reporting as listed below:

1. VAF distribution queries.

1a. In Fig 1b -- In 1 of the 3 shown patients, VAFs tend to be centered at ~0.35-0.4, quite consistently. What could be causing that?

1b. Related, VAF distributions of some (few) microdissections appear to result from a mix of clones. These could be easily identified by e.g. a mixture model or such. This might influence technical factors in bioinformatics pipelines. Do they present as outliers in the analysis associating mutation burden with age (or with H.pylori), or in the mutational signature analysis?

2. Statistical consideration of how missing data affects interpretation of analyses.

2a. Lines 130-135. „Pathology review of hypermutant glands in the antrum revealed that six out of seven donors exhibited local chronic inflammation“ What proportion of non-hypermutant glands exhibited local chronic inflammation by the same criteria?

2b. Line 137. „Annotated current or previous H. pylori status, where known, did not significantly affect SNV burdens (p=0.07...“ p=0.07 probably favors that the association of Hpy to mutation rates exists rather than the converse, and I think this paragraph might be understood differently than they intended. If there really were past, undetected infections with effects on mutation rates, this association would be biased conservatively. Probably needs rewording. Also check association with individual mutational signatures, if not checked already.

3. I have several worries about the mutational signature analysis:

3a. Lines 196-202 describe SBS mutsig in hypermutant glands. Increase in the ageing-associated SBS1 and SBS5/40 (1.7-4 fold) is smaller than the increase in non-ageing-associated (to my knowledge) SBS18 (11-fold), but they conclude the hypermutation is due to increased proliferation of stem cells? This does not quite fit. Present additional evidence and/or clarify, or remove „it is plausible“ claim.

3b. Regarding ID1/ID2 ratio being significantly changed in hypermutating glands – again it is not clear how this links to increased proliferation (lines 210-212). Is ID2 known to be more strongly proliferation associated with ID1 – where is the data/reference to support that? In absence thereof the claim does not appear to stand.

3c. Line 630: „As ID1 and ID2 are highly specific signatures, their contribution was estimated based on the number of single base insertions and deletions of homopolymer runs of A and T of length six or greater, respectively.“ Justify why for indels the same procedure was not used as for SNVs? (based on mSigHdp + postprocessing).

3d. Line 226. „The pattern of mutational signatures in glands dissected from the two gastric cancers was markedly different from that in normal glands.“ Please state the proportion of signatures in the text (ideally do a statistical test, although sample size might preclude this). APOBEC contribution does not seem major (in line with APOBEC mutagenesis not being that highly prevalent in stomach cancers).

4. Some concerns regarding driver alterations

4a. They seem to comment only on the SNV in this context, but not on the CNV. Are the 3 types of CNVs (focal, arm level ccn-loh, and whole-chromosome events) statistically enriched with known dosage-sensitive oncogenes, or TSGs? (that there is an enrichment in fragile sites does not preclude there is also an enrichment in driver events) Do they observe two-hit events with the CNV, e.g. possibly some germline variants might cause selection on the recurrent CNV within 1 individual?

4b. Driver mutations and age „The average number of driver mutations per gland per individual correlated with age (p=0.008“ does this association remain when controlling for general increase of mutations with age?

4c. Potentially important concern about method to detect driver variants. Line 647: „We used the dndscv24 R package to identify genes under positive selection, combining both the whole-genome sequencing data and the targeted sequencing data“ It is not clear how the 2 data types can be combined here: the modelling of mutation risk from covariates in dNdScv would not be expected to work from targeted sequencing data. Can they explain better how did they go about this? Can they demonstrate with data that dNdScv is safe to use in this way (mixing WES and panel sequencing, if they did so)?

5. Bioinformatics question. Line 578: „The truncated distribution is necessary to reflect the minimum number of reads that support a variant ($n = 4$) that is imposed by variant callers such CaVEMan“ My intuition is that forcing 4 reads to support a variant may be quite stringent, and aggravates the confounding by sequencing depth, which then necessitates they have to correct for it (lines 591-597) introducing extra complexity and possibly errors.

6. Data availability. Please make the somatic mutation calls available (not only raw data/reads), preferably without restrictions, to facilitate reproducibility and reuse.

7. Methods text could be improved by edits. Below are suggestions for some clarifications/corrections/added references. There could be others; please go over methods with a fine-tooth comb and correct errors and ambiguities and citations.

Line 575-576 „For every indel or SNV, the overdispersion parameter (ρ) was determined in a grid-based way (ranging the value of ρ from 10^{-6} to $10^{-0.05}$)“ is confusing/incomplete. „Grid-based“ but only one parameter is varied (not two or more)? What is the criteria for choosing best ρ ?

Line 602: „we included site-specific age relations in the mixed effects model“ vague – I infer this is an interaction term age:site? Please include all details necessary to understand and reproduce analysis.

Line 611: „To identify possibly undiscovered mutational signatures in human placenta“ Placenta? Are the other parts of the Method also copy-pasted from another text without checking well if they apply here? Please check.

8. Further minor remarks.

SBS17 signature consists of A>C (T>G) mutations, which do have reported links with acid exposure (in a model system) and oxidation of the free nucleotide pool. This seems a more relevant explanation than the 5-FU association of SBS17 that is mentioned.

Line 165: Does SBS18 have to result from *endogenous* ROS?

Line 567-569: „Germline variants were removed using a one-sided binomial exact test on the number of variant reads and depth present across largely diploid samples, as previously described.“ Seems to lack a reference.

Line 523: „Custom Agilent SureSelect bait set capturing the exonic regions of the following 321 cancer associated genes:“ please put them in a supp table instead of listing them here, which is distracting.

Discussion is quite long.

Version 1:

Reviewer comments:

Referee #1

(Remarks to the Author)

The authors have provided in depth responses to my concerns and suggestions. The manuscript is much improved. I do not have additional concerns and I look forward to seeing it in print. I expect the work to be well read and cited.

Marnix Jansen, UCL

Referee #2

(Remarks to the Author)

I would like to thank the authors for their thorough responses to my comments and congratulate them on this elegant work.

(Remarks on code availability)

A section on how to install in the README (including a list of dependencies), would be useful for users who would like to reproduce the analyses.

Referee #3

(Remarks to the Author)

In the revised study, Coorens et al. have satisfactorily addressed my comments and those of other reviewers by refining various genomics analyses and statistical analyses, and by including more systematic pathology assessments. These revisions make the reported data more robust, and the presentation has also improved. The study contains highly relevant, timely findings for the somatic evolution field and is also likely to be of substantial interest to a broader audience.

The somatic mutation landscape of normal gastric epithelium (2024-03-04772)

Response to reviewers' comments

We would like to thank the reviewers for their helpful suggestions and comments on our manuscript. We have detailed our response to each point below, with the original comment in black and our response in blue text. There are three main aspects to this revision:

1. We now include the pathology annotation of all gastric microdissections, for both whole-genomes and panel sequencing data, which allows us to draw more substantial conclusions on the mutational landscape of metaplastic versus “normal” gastric glands. To maximize the clarity and clinical relevance of the manuscript, we have dropped the term genomics-centered “hypermutant” gland and replaced it with by focusing on metaplastic glands, as suggested by Referee #1. The conclusions are in line with the previous version: metaplastic glands have increased SNV burdens (due to SBS1 and SBS18), increased indel burdens (due to ID1 and ID2) and increased intrachromosomal SV burdens.
2. We have performed additional analyses as requested by the reviewers, including indel mutational signature extraction, telomere length estimation, a more in-depth analysis of SVs, an updated driver analysis, and a more thorough analysis of the clonality of gastric glands.
3. We have ensured the underlying data, scripts, and methods for the study are available to the wider public. Raw sequencing data has been deposited in the European Genome-Phenome Archive under controlled access. All scripts for analysis and intermediate data files used for the analysis, such as filtered mutation calls, phylogenetic tree files, signature data, have been deposited in the GitHub repository accompanying the paper (<https://github.com/TimCoorens/stomach>), which now features a detailed description of its contents.

Referee #1:

In this manuscript Coorens and co-workers describe their work analysing the somatic mutation type and burden of normal and (pre)malignant gastric glands. The methods follow an approach which is now well described focusing on ~30x WGS of microdissected individual clonal units in the luminal gut. The data are well-presented and I particularly like the way this study is framed against the background of previous studies from this group examining mutation accumulation in the oesophageal, small intestinal and large bowel epithelial lining. I feel this is an excellent piece of work.

I have two major points of critique and several minor points of concern for the authors' consideration. My two main points of critique focus on so-called hypermutant glands (1) and the

polyclonal microbiopsies which have been analysed through targeted capture sequencing (2). I will address these in turn, with apologies for the (uncharacteristic) lengthy review.

1.1 Hypermutant glands: Here the authors make the case for a select group of glands which have gone through a period of accelerated mutation accumulation (as per line 125-127: ‘it appears that each gland has independently increased its mutation rate during the lifetime of the individual in response to a local stomach environment in which a gastric cancer has developed’). These data overall are very persuasive – a population of glands clearly stands out, both in terms of mutation burden (Fig 2) and in terms of mutation signature distribution (Fig 3e). These glands are mainly found in the antrum (nb. although the antrum is regrettably under-sampled in the non-cancer donors as per Fig 2, top panel) and are specifically associated with cancer donors. Importantly, it appears that nearly all hypermutant glands (except for one) derived from intestinal metaplasia. This latter point could have major implications for our understanding of gastric cancer progression, but I fear this message might be lost because of the semantics (see below).

Response: We thank the reviewer for highlighting this concern and have changed the flow and presentation of the manuscript accordingly, to focus on pathological differences (inflammation and metaplasia) that show differences in mutational landscape, rather than the other way around. To this end, we now include pathology grading for metaplasia and inflammation for all WGS microdissections and biopsies (listed in Extended Data Table 2). We find that microdissections from metaplastic glands have a significantly higher mutation burden, largely fuelled by increases in SBS1, SBS18 for SNVs, ID1 and ID2 indels, and an increase in the number of intrachromosomal structural variants. Thus, the previously highlighted relationship between intestinal metaplasia and hypermutation strongly holds. During the revisions we have adjusted Figure 1, Figure 2 and the accompanying text. We further point out the more detailed responses to the reviewer’s sub-points below.

Published evidence shows that intestinal metaplasia is a clonal precursor (PMID: 18242216) which progresses to dysplasia (PMID: 21223968). Although the authors suggest a temporal increase in mutation *rate* (my emphasis) to explain the increased mutation burden in these glands, no mechanistic evidence for this is provided. By contrast, the authors also state (line 200-202) that ‘since SBS1 and SBS18 mutation rates have previously been associated with increased cell division rates and inflammation, it is plausible that the hypermutation in these glands reflects periods of increased stem cell proliferation’. This would instead suggest that the increased burden is not due to an increased mutation *rate*, but rather secondary to an increased cell turnover.

Response: We agree with the reviewer that the increased mutation burden is likely (at least in part) caused by an increase in cell proliferation. When alluding to a mutation rate, we generally

mean the number of mutations accumulated per year, rather than per cell division, as it is straightforward to incorporate age in our calculations, whereas the number of cell divisions are not directly observed. We have elaborated upon this point in the Discussion section to hopefully further clarify our interpretation of the data:

“Gastric glands with intestinal metaplasia, which are often associated with chronic inflammation and local clonal expansions, show increases in total mutation burdens due to elevated SBS1 (methylcytosine deamination), SBS18 (reactive oxygen species), ID1 and ID2 (replication strand slippage) and intrachromosomal SV mutation rates. These changes in mutagenesis could reflect an increase in cell division rates in metaplastic glands, be the consequence of other factors intrinsic to such cells or be due to microenvironmental influences.”

Evidence indicates that gastric glands undergo niche turnover through a process of neutral competition akin to colonic crypts, although details as to number of stem cells and clearance rates in gastric stem cell niches remain unclear. In *Helicobacter*-infected gastric epithelia often intense cytotoxic cell death is seen around the gastric isthmus stem cell zone and stem cell proliferation rates are clearly increased compared to uninfected gastric mucosa. Ultimately, this drives somatic selection and expansion of metaplastic lineages which have gone through a more recent bottleneck compared to any randomly-selected background gastric gland.

Overall, the most parsimonious explanation for the observed increased mutation burden in this ‘hypermutant’ category of glands is not increased mutation *rates*, but rather more likely decades-long accelerated turnover in a chronically inflamed environment coupled with clonal selection and expansion of metaplastic lineages. This interpretation of the genetic data in this study is akin to the two-step initiation-promotion process first described in mouse models and recently revived by the Swanton team (see PMID: 37020004 and PMID: 37059064) in the context of air pollution-linked EGFR mutation in chronically inflamed lung epithelia. In this model the first step is stochastic acquisition of an oncogenic mutation, often through endogenous clock-like mutational processes, and the second is exposure to an inflammatory stimulator, which drives lineage selection and expansion, in particular of those adapted to the inflammatory environment. The two steps together are required for tumour formation. These discoveries have important implications for preventative medicines targeting the first steps in tumour evolution. The authors’ approach to interrogate mechanisms of early cancer initiation in normal tissues is ideally suited to address this point and would make a highly impactful public health message, namely that gastric cancer initiation is due to chronic inflammation-associated accumulation of endogenous clock-like oncogenic processes in clonally selected lineages.

Response: We have now incorporated our findings and interpretation on the role of metaplasia and inflammation in tumor promotion in the Discussion section. With pathology annotation now available for all microdissections, we have evaluated the role of chronic inflammation and

intestinal metaplasia on mutation burden, telomere length and selection landscape. We find a strong effect of intestinal metaplasia on the burden of SNVs, indels and intrachromosomal SVs across samples, and an effect of chronic inflammation on telomere shortening, the occurrence of independent trisomic gains and the proportion of mutant epithelium, especially severe chronic inflammation.

“Severe chronic inflammation was significantly associated with elevated numbers of driver mutations in gastric glands and overall proportions of mutant epithelium in this study, highlighting a role for chronic inflammation in molding the pre-neoplastic selection landscape, as also identified in inflammatory bowel disease³⁰. Beyond a role for inflammation, the large variation in this proportion across donors may indicate between-donor variation in selective pressures. Larger studies may be powered to find exposures or other agents that further impose the selection of specific clones, as has been found for smoking in oesophagus⁴, and promote the transformation of normal cell to overt tumours via metaplasia and dysplasia.”

In this context I fear that ‘hypermutable’ glands suggests some burst-like mutational process which distracts from the underlying clonal dynamics. On a tangent to this, the authors also state that (line 124) ‘Hypermutable glands were phylogenetically unrelated to each other beyond early development, except for one clonal expansion detected in PD42789 (Extended Data Fig. 1)’. It is unclear whether this statement rules out that clonal expansion in hypermutable lineages is limited to isolated examples (like PD42789), as this would require additional dense gland-by-gland sampling around an IM gland (low power images are not provided to assess this). I would suggest that the clonal expansion shown in Extended data 1c instead indicates that intestinal metaplasia can undergo sizeable clonal expansions (in line with PMID: 18242216).

Response: We thank the reviewer for suggesting to incorporate more analysis of clonal dynamics in metaplastic glands. Our sampling strategy (as now exemplified by more high-level histology overviews in **Extended Data Figure 2**) remains sparse. While it indicates that metaplastic glands sampled far from each other are independent clones, we do see clonal relations for microdissections of metaplastic glands sampled closely (e.g. PD40295 and PD42789). In addition, we now show that the median VAF, a measure of the clonality within a microdissection, is significantly higher for metaplastic glands compared to non-metaplastic glands ($p=0.006$, Extended Data Figure 1b).

We have added the following to the text:

“Metaplastic glands exhibited overall higher median VAFs per microdissection compared to non-metaplastic glands ($p=0.006$, Wilcoxon rank sum test, **Extended Data Fig. 1b**), and were closely related phylogenetically when anatomically close to each other (**Extended Data Fig. 2a**), suggesting that metaplastic clones locally expand. However, metaplastic glands from the same

donor at a distance from each other were phylogenetically unrelated beyond early development (Extended Data Fig. 2b,c), indicating a wider “field effect” of metaplasia induction.”

To summarise I would suggest the authors revisit this description and preferably drop the potentially confusing term ‘hypermutant’ throughout. Instead, to drive mechanistic and clinical impact I would suggest the authors specifically label the metaplastic samples in Figure 1B, Figure 1C to F and Figure 2, and focus the discussion of their data on the accumulation of clock-like mutational processes in chronically inflamed epithelia (likely *Helicobacter*-associated) and eventual selection and expansion of intestinal metaplasia. Direct in situ demonstration of the clonal initiation of intestinal metaplasia would make a valuable addition to the manuscript.

Response: We thank the reviewer for the in-depth comment. As outlined, we have rewritten the manuscript to focus on metaplastic rather than “hypermutant” glands throughout and focus on interpreting the relation of mutational patterns with histology observations (metaplasia and chronic inflammation). We hope the revised manuscript now adequately uses the powerful combination of genomic data and pathology to shed light on the pre-neoplastic somatic evolution in the gastric epithelium.

1.2 Targeted capture data: Unlike the colonic mucosa which essentially consists of a monotonous repetition of clonal units, clearly spatially separated, regardless of the plane of sectioning, the gastric mucosa has a much more complex arrangement characterised by clonal units which extensively branch towards the base. Earlier studies using classic sporadic marker alleles have shown that these branches are clonally related and therefore aren’t individual units (see for example PMID: 8062232). An important consequence of this is that the cellular progeny of a stem cell carrying a somatic variant likely streams into multiple individual glands/tubules.

Although it isn’t spelled out directly, the wording in the manuscript suggests that each ‘gland’ (that is, a branch/tubule of a clonal unit) is counted as one clonal unit akin to a colonic crypt. Further complicating this issue is that individual glandular branches often mingle. In this way, each of the ±800 microbiopsies contains on average 10 glands, but those 10 glands could all derive from a single stem cell niche or (more likely) from multiple neighbouring stem cell niches. These spatial considerations of course affect the interpretation of spatially clustered mutations, variant allele frequencies and modeling data.

Response: We thank the reviewer for pointing this out and have now included a more detailed description of the clonality of the gastric glands, as determined by their median VAF (Fig. 1c), as well as the accompanying text:

“The clonal composition of gastric glands can be estimated from the variant allele fractions (VAFs) of somatic single nucleotide variants (SNVs) and small insertions and deletions (indels). The

median VAF per microdissection generally exceeded 0.25 (Fig. 1b,c), which – allowing for a degree of stromal contamination – confirms the notion that most glands are dominated by the progeny of a single stem cell, a clone which takes up more than half of all cells. In 8% of microdissections (17/217), the median VAF was below 0.25, or had evidence of multiple clones co-existing, suggesting a continued presence of multiple stem cell niches and hierarchies more complicated than those observed in intestinal crypts⁵.”

In addition, we have added the median VAF for the panel genome sequencing samples to **Extended Data Table 3**.

1.2a My primary concern is that without some introduction to these caveats that the general reader may view these data to mean that ‘glands’ correspond with individual clonal units in the ‘crypt’ sense. I would suggest to (briefly) introduce these concepts and avoid the term ‘gland’ in this context since its meaning is ambiguous, in particular when analysing polyclonal input. For example, where the abstract states ‘Surveying approximately 8,000 gastric glands by targeted sequencing’ this could be misinterpreted as 8,000 individual clonal units. I would instead suggest ‘Surveying approximately 829 polyclonal gastric microbiopsies by targeted sequencing’ (cf. line 291 and Fig 5c).

Response: We have made the suggested change, and now report the total area surveyed per donor in Fig. 5c. Rather than using the estimate of the number of glands per microdissection as a basis for our calculation of the proportion of gastric glands with a driver mutation, we now use the estimated microdissected area as input. The response to point 1.2c goes into greater detail on the precise methodology used.

1.2b This mingling of glands and clonal units likely explains the spatial clustering of KDM6A and ARID1A variants (greater curve biopsy, Fig 5D top) and one of the beta-catenin variants (lesser curve, Fig 5D bottom). By extension, the maximum parsimony phylogeny shown in Fig 5D therefore isn’t necessarily incorrect, but its meaning is different from the beautiful phylogenies in Fig 3. Again I feel this must be highlighted.

Response: We apologise for the lack of clarity. The phylogeny shown in Fig. 5d is based on the WGS data of microdissections of single glands in this patient, while the histology images show both the WGS and panel sequencing data. We have revised the figure legend to improve clarity: *“d, Phylogenetic tree of donor PD42790 (80, M) from the whole-genome sequencing data of individual glands, annotated with putative driver mutations, along with histology images from two regions overlaid with driver mutations and their VAF, of both whole-genome sequenced microdissections (indicated by WGS) and panel sequenced clusters of microdissections.”*

1.2c I suspect this spatial clustering effect also affects the calculation shown in Ext Data 7 (relation between age and percentage of gastric glands with drivers) and its interpretation ('in 60-year-old individuals approximately 5% of glands were colonised by clones with drivers'). The assumptions that underpin the model must be made clear.

Response: To avoid the aforementioned assumptions that microdissected glands are all independent, we have now used the area of each microdissection in our calculation of the driver prevalence across the gastric epithelium.

To estimate the proportion of gastric epithelium that harboured a driver mutation can then be estimated from the measurements of area of microdissections and the cell fraction that harbors a driver mutation within a sample. This cell fraction is straightforwardly obtained by multiplying the VAF of a mutation by the local ploidy (1 for sex chromosomes in male donors, 2 otherwise). Multiplying mutant cell fractions with the area of epithelium sampled gives an estimate of the area of mutant epithelium. By summing up the total area of mutant epithelium and dividing by the total epithelial area sampled we arrive at an estimate of the fraction of gastric epithelium that has been colonized by a clone with a driver mutation. For this analysis, we used both disrupting (missense, nonsense, frameshift and splicing) mutations in genes identified as under selection by dNdScv and manually annotated driver mutations (see previous section).

Note that this approach assumes that the entire microdissected area consists of gastric epithelium. Any contaminant cell type will lower the VAF of mutations in gastric clones and therefore reduce the estimated mutant sizes. Therefore, the estimated proportion of epithelium can be slightly underestimated.

The results are presented in **Extended Data Figure 9**.

1.2d Finally, in a similar vein I suspect that since the dNdS calculations were derived by combining single gland WGS and targeted capture data from microbiopsies (Methods) that spatial mingling of clones and spurious recurrence of variants could also skew these data. This must be addressed.

Response: The dN/dS calculations are based on counting a certain mutation once per individual, to avoid assuming the presence of a certain mutation in multiple microdissections was due to independent mutations, as the latter scenario would indeed skew the results through spurious recurrence. Hence, this approach is robust against spatial mingling of clones in neighbouring microdissections.

As part of a wider revision to the dN/dS calculations (see also response to point **4.3c**), we have greatly expanded the Methods description of the dN/dS analysis, including the consideration of branching structure of gastric glands and avoiding double counting of mutations. In the Methods

section, we have added: “To avoid overestimating the occurrence of mutations because of sampling the same clone in multiple microdissections, we only count a specific mutation once per donor.”

Overall I feel that these targeted capture data on polyclonal units must be presented with the proper caveats.

We thank the reviewer for the expansive set of comments related to the clonality of gastric glands and sampling, and hope we have adequately incorporated the feedback.

Further points:

1.3 I suspect that the data in Fig 1B are from individual glands only, not patient-matched microbiopsies. I would suggest emphasising this point.

The data we show are the VAFs of mutations found in individual microdissections of single gastric glands per donor. We have clarified this in the legend: “VAF distributions of somatic mutations in gastric gland **microdissections** for three donors, coloured by the median VAF.”

1.4 Line 279/280: ‘Glands with trisomies were not enriched in individuals with gastric cancer ($p=0.48$, Fisher’s exact test) and were not significantly more likely to harbour driver mutations than glands without trisomies ($p=0.74$, Fisher’s exact test).’ The authors do not provide a dedicated discussion on driver mutation data from the 30x WGS gland set. This feels like a gap in the story and I would suggest this is addressed.

Response: We have added driver mutation annotation for all WGS samples to the phylogenies in **Extended Data Fig. 3** and **Extended Data Fig. 4**. Further, we now analyse the occurrence of driver mutations with respect to intestinal metaplasia and chronic inflammation, and find that glands with severe chronic inflammation are significantly enriched for driver mutations. We now discuss this in the text:

“Driver mutations in the whole-genome sequencing data were largely confined to single microdissections (**Extended Data Fig. 3** and **Extended Data Fig. 4**) and while there was no enrichment of drivers in metaplastic glands ($p=1$), glands with severe chronic inflammation were significantly enriched for drivers ($p=0.01$, Fisher’s exact tests).”

1.5 Line 285: ‘However, had a pathogen been implicated in trisomy acquisition, the infection may have predated the sampling by decades and would be unlikely to have persisted at readily detectable levels at the time of biopsy.’ It is thought that most *Helicobacter*-positive patients are infected early in life (<10 yo) and that infection is maintained throughout life unless a patient is either treated or significant atrophy sets in which means *Helicobacter* can no longer thrive (*Helicobacter* autolyses at $pH>5$). Without direct serologic information, atrophy status or prior

treatment status it is not possible to state that it is unlikely to have persisted and this statement must be toned down or qualified.

Response: We have revised this section. The annotation of all inflammation status of all glands (as suggested by the reviewer), allowed us to formally test the association between inflammation, metaplasia and trisomy acquisition. We found a significant association with moderate/severe chronic inflammation and the burden of trisomies ($p=0.003$), which prompted us to change the text and interpretation to:

“The cause of this distinctive pattern of CNVs in gastric glands is uncertain. In PD41767, trisomies are detected in four of ten glands in one stomach biopsy of the antrum, but wholly absent from seven glands sampled elsewhere in the antrum. Both age and the presence of metaplasia have a significant effect on the burden of intrachromosomal CNVs ($p=0.01$ and $p=10^{-14}$, respectively, ANOVA test). While the burden of trisomies is not significantly associated with age ($p=0.38$), metaplasia ($p=0.84$), or whether the donor had gastric cancer ($p=0.63$), there is a significant association with severe chronic inflammation ($p=0.003$, all tests are ANOVA tests).

Our data suggests that, rather than a continuous age-associated increase of whole-chromosome duplications, these trisomies were generated at a specific time during the lifespan of each individual, and possibly confined to specific regions of the stomach. The process of acquisition or selection of these trisomies is not apparently linked with metaplasia or carcinogenesis, but there is possibly a link with chronic inflammation. While none of the donors harbouring trisomies were known to be infected with *H. pylori*, the confinement in space and time, and the association with inflammation are suggestive of the involvement of an exposure or infection with a pathogen.”

1.6 Extended data figure 1 – the panels carry helpful histopathology annotations with regards the presence of chronic inflammation and grade of intestinal metaplasia. As stated before, I feel additional lower power overviews may be instructive to reveal the spatial density of sampling, also with regards to statements on the phylogenetic relatedness/clonal expansion of metaplastic clones. Second, international guidelines recommend that the extent of intestinal metaplasia across the stomach is graded, not the degree of involvement within a single gland.

Response: We have added additional lower power overviews to this figure (now **Extended Data Figure 2**). We apologize for the confusion of the metaplasia grading and now provide grading per gland and grading per region in **Extended Data Table 2**.

Referee #2 (Remarks to the Author):

Tim Coorens and colleagues propose an impressive dataset of 238 whole-genome and 829 targeted sequencing of laser-microdissected gastric glands across 30 donors, including 18 cancer donors.

After similar studies on the human oesophagus and intestines, this is the first large study of the somatic evolution of the human stomach in the non-cancer and cancer contexts.

They focus on the evolution of 'normal' non-cancer glands in those contexts and contrast with 21 cancer microdissections.

First, most glands were monoclonal with VAF of the mutations close to 0.5 (>0.25). Akin to other organs, they show that the mutation rates increases linearly with age at a rate of 27.8 SNVs and 2 indels per year. Some glands, mostly in the antrum of 7 cancer donors, showed significant excess of SNVs relative to their age-related baseline, which they refer to as hypermutant glands. These hypermutators were unrelated phylogenetically, suggesting a common hypermutator environmental rather than genetic background. Indeed, most of these cases turned out to have chronic inflammation and/or metaplasia, while one case had no inflammation nor metaplasia. More generally, the SNV/indel burden in glands of cancer donors was significantly higher.

The mutations in normal glands followed patterns of known mutational processes, including clock-like (SBS1, SBS5/40), and ROS (SBS18). SBS1/5/50/18 were present in all glands, suggesting generic processes. Clocklike mutations accumulated linearly with age but not SBS18. The proportions of signatures related to replication, SBS1 and SBS18, increased in hypermutators, suggesting periods of higher proliferation leading to those. Indel proportion also increased with signatures linked to errors during replication. Whether this is due to metaplasia leading or in response to cancer is unclear. SBS17ab, which is rarely seen in normal tissues, but was seen in oesophagus and gastric cancers or Barrett's oesophagus, was present in a few normal glands of cancer donors, suggesting processes generating them in a normal context in the stomach. APOBEC signatures (SBS2/13) were also not seen in normal glands, unlike in gastric cancers.

Interestingly, unlike in other normal human tissues, CNVs were not rare, mostly smaller events at fragile sites, chromosome arm events leading to *cnn*-LOH, and trisomies of chr20 and chr13 in normal gastric glands, and could arise independently in glands from different branches of the phylogenies within the same donor. Thanks to gain-timing analyses, the authors show that gains of chr20, akin to its timing in gastric cancers, arose early in the patients lifetimes. They were not

linked to cancer patients, a specific anatomical location, or the presence of driver mutations, but related to hypermutators.

Next, the authors profiled 321 known cancer genes with high-depth targeted sequencing in an extra 834 microdissections, each representing multiple glands. They found evidence of positive selection in 6 genes: ARID1A/B, CTNNB1, KDM6A, ERBB3, and LIPF, a gastric lipase. Except for LIPF, all were identified in gastric cancers. However, CTNNB1 had patterns reminiscent of tumour suppressor in normal glands rather than oncogene in cancer glands. Some of the driver patterns (e.g. one CTNNB1 mutation) showed converging evolution confined to mm space. Other likely drivers at known hotspot locations were found, e.g. BRAF, but not in TP53 or PIK3CA, two common drivers in gastric cancer. The number of drivers increased with age such that by age 60yo, 5% of the glands had a driver mutation.

Altogether, this is an important novel study for the field of human somatic evolution, describing the somatic evolution of the human stomach for the first time, across age groups and cancer vs. non-cancer donors. It is a purely genomics-based study, repeating analyses already performed in other organs and with no molecular phenotypic data. The originality stems from the exclusive dataset for the human stomach rather than the now well-established methods. I believe it is an important high-impact work that deserves publication.

It would be great if the authors could answer a few major comments and some minor comments for this work. Please see below.

Major Comments.

2.1 Line 685. Data availability. Please make sure to publish the data.

Response: The manuscript now contains the following data availability statement:

DNA sequencing data have been deposited in the European Genome-Phenome Archive (EGA) with accession codes EGAD00001015351 (whole-genome sequencing) and EGAD00001015352 (targeted panel sequencing). Processed data are available in the Extended Data Tables or on GitHub (filtered variant calls, phylogenies, etc.).

2.2 Line 681. Code availability. The github code should have a README with a clear description, without which it is unnecessarily challenging to reproduce the analyses.

Response: We have greatly extended the extent of the code deposited on the GitHub, including a README, input data and expected output.

2.3 As acknowledged by the authors, there is no antrum data in the non-cancer donors. But I do not follow the reasoning in the discussion. What is a "diseased antrum"? Do the authors mean that these hypermutant glands would be exclusively seen in the cancer context, or in a metaplasia/inflammation context? To me it could very well be that the antrum in the absence of cancer generally has regions with glands of high mutation load, potentially linked to metaplasia/inflammation. If this was the case, I do not think the current design would have allowed to see it. While the authors talk about an 'association' with metaplasia/inflammation, they do not show metaplasia/inflammation for the whole cohort, and do not show it is absent in non-cancer donors. I think the authors, if they could not add antrum data in the non-cancer donor group, should at least show/discuss the metaplasia/inflammation status in the other cancer donors and the non-cancer donors, which to my knowledge could very well be positive in a large proportion.

Response: We thank the reviewer for this comment, and it echoes remarks made by the other reviewers. We have now included pathology annotation for all whole-genome sequenced microdissections in the study, both for chronic inflammation and intestinal metaplasia. While chronic inflammation is widespread in our donor cohort (even in non-cancer donors), we only observed metaplasia in gastric antrum of cancer donors. Strikingly, metaplasia (either complete or incomplete) is very strongly associated with an increased mutation burden even when confining to samples from the gastric antrum in cancer donors. This increase is strongly evident across classes of somatic mutation: SNVs, indels and intrachromosomal SVs.

Other comments.

2.4 Line 86. "The mean variant allele fractions (VAFs) of somatic single nucleotide variants (SNVs) and small insertions and deletions (indels) generally exceeded 0.25 (Fig. 1b), indicating that gastric glands are predominantly monoclonal cell populations derived from recent single stem cell progenitors." Could the authors please detail the reasoning a bit more, i.e. on the rationale of the 0.25 threshold and its relation to monoclonality and the recency of the stem cell progenitor.

Response: A mean VAF of over 0.25 indicates there is a clone occupying over 50% of the microdissection (because most somatic mutations are heterozygous). No two clones can each occupy more than 50%, which means the high VAF indicates most microdissection clones are indeed the progeny of a single (stem) cell. Estimating the recency of the stem cell progenitor requires a few more assumptions. From the linear mixed effects modeling, we estimate that gastric glands accrue ~28 SNVs per year, with an intercept of ~56 SNVs. The intercept represents the mutation burden at birth minus the time it takes for the stem cell to colonize the microdissection. Estimates of mutation burden in gastric glands at birth are currently lacking, but neonatal (umbilical cord) blood carries approximately 55 SNVs at birth (Mitchell et al., Nature,

2022) and it is likely the burden in gastric glands at birth is not far from cord blood burden. The closeness of the estimated intercept and assumed mutation burden at birth would indicate that gastric stem cells colonize the gland fast, likely a timescale much shorter than a year. This notion is further confirmed by studies supporting a quick epithelial cell turnover rate in the stomach (days) and high rates of stem cell division (a division every 5 days). (<https://www.sciencedirect.com/science/article/pii/S2211124713005482>)

2.5 Fig 1cd. the n value is next to donors rather than glands, which is inaccurate.

Response: We have fixed this to ensure it is clear the “n” refers to numbers of glands.

2.6 Fig 1cdef. hypermutators outside 95% interval could already be highlighted (e.g. in red) to help with figure glancing.

Response: These figures have been changed in the switch from a genomics-based view (“hypermutant”) to a pathology-based analysis now presented in Fig. 2 (“gastric site/inflammation/metaplasia”).

2.7 Fig 1g. the label of the two groups under number should be something like "Glands, non-cancer donors", as otherwise the numbers seem to report to donors rather than glands again.

Response: We have fixed this to ensure it is clear the “n” refers to numbers of microdissections of single glands.

2.8 Ext Fig 2bc, Did you count only clonal mutations in the tumour microdissections - could it be that a large fraction of the excess is simply explained by later subclonal mutational events?

Response: We have added the clonal tumour mutation burden as a separate dot in the scatter plot (black) in Extended Data Fig. 1d,e (note the change in numbering). In PD41759, an average of 95% of SNVs and 97% of indels were clonal; in PD41762, this was 86% of SNVs and 91% of indels. Hence, the excess of mutations is not explained by later subclonal events, but rather hypermutation in the trunk of the tumour. This can also be seen in the tumour phylogenies in Fig 3c,d.

2.9 Fig3abc. We suggest removing border in the bars, such that the colour of smaller contributions still have a chance to appear (currently overshadowed by the black borders).

Response: We have made the borders in the bar graphs much narrower to allow smaller contributions to be visible (**Fig. 3a-c, Extended Data Figure 3 and Extended Data Figure 4**). The full contributions of each signature to each microdissection are also reported in **Extended Data Table 2**.

2.10 Regarding the relation of SBS18 with age. In the phylogenies where two glands originate from a "late" recent common ancestor, SBS18 mutations are mostly seen in the late branches, even in the relatively later branching in mutational time. This would contradict the hypothesis of early development processes. Could the authors comment on this. Related to this, could the authors also use the mutations at multiplicity 2 (vs. multiplicity 1) on trisomies to time SBS18 signatures in the early molecular-time periods?

Response: With the shift in focus on pathology annotation (metaplasia), we re-analysed the age-related burden for the three ubiquitous mutational signatures (SBS1, SBS5/40 and SBS18). It is now significantly associated with age (**p=0.047, Extended Data Fig. 5f**). We have updated the text accordingly. The clock-like behaviour for SBS18 has also been reported in the colon (Lee-Six et al., 2019, *Nature*) and small intestine (Wang et al., 2023, *Nat. Genet.*).

2.11 Line 200. "Since SBS1 and SBS18 mutation rates have previously been associated with increased cell division rates and inflammation". Please provide a citation for this.

Response: We have removed this sentence from the manuscript and interpret the elevation of mutation burdens in metaplastic glands in the discussion, including specifically:

"Large intestine epithelial cells in areas affected by the inflammatory bowel diseases Crohn's and ulcerative colitis also show elevated mutation burdens with increased proportions of SBS18 mutations³⁰."

2.12 Fig 4b. This figure in a table format should also show information of cancer vs. non-cancer donors.

Response: We have added the cancer/non-cancer annotation to both **Fig. 4b** and **Fig. 5b**.

2.13 Line 273. "There was no significant effect of age on the burden of trisomies." This seems surprising from the table on Figure 4b, where clearly trisomies are enriched in older patients within sex-groups. Are CNVs more present in cancer donors? Could it be that the statistical test used is not powered to detect the enrichment clearly present in this table? Could it be most patients acquire CNV early in life, which are typically selected for later in life?

Response: We have expanded this sentence and analysis to leverage the full annotation of inflammation and metaplasia and find that, while not associated with age, metaplasia or cancer status, the burden of trisomies was associated with inflammation. In the text:

"While the burden of trisomies is not significantly linearly associated with age ($p=0.38$), metaplasia ($p=0.84$), or whether the donor had gastric cancer ($p=0.63$), there is a significant association with severe chronic inflammation ($p=0.004$, all tests are ANOVA tests)." In addition, we have added a dedicated figure on the age estimates of the chromosomal gains (**Extended Data Fig. 7**), which shows many of these gains are acquired early in life. Later selection of earlier

gains is definitely possible, but it does not explain the temporal clustering of trisomic events, especially in PD40293.

2.14 Line 282. "However, there was a modest enrichment of trisomies in hypermutated glands ($p=0.03$, Fisher's exact test), suggesting a role of factors causing chronic inflammation in their genesis." Why inflammation and not metaplasia instead?

Response: With the inclusion of inflammation and metaplasia annotation for all samples, we have updated this analysis and the accompanying description:

"While the burden of trisomies is not significantly associated with age ($p=0.38$), metaplasia ($p=0.84$), or whether the donor had gastric cancer ($p=0.63$), there is a significant association with severe chronic inflammation ($p=0.003$, all tests are ANOVA tests)."

2.15 Line 329. That there are 5% glands with drivers at 60yo: this is a useful number but it should come with a sense of the biological/technical variability. The confidence interval on this estimate is very large and the observed values for the few donors around 60yo range span an order of magnitude (from 1% to 10% ; extended data fig. 7). It does not seem like there is a sharp shift in the elderly like for the clonal diversity in haematopoiesis. Could the authors also comment on this variability and how it compares to other organs and segments of the gastrointestinal tract.

Response: We have included an analysis to test the influence of metaplasia or inflammation on the proportion of the gastric epithelium with driver mutations, and find a significant effect for chronic inflammation on the proportion of driver clones.

"The proportion of the gastric epithelium, surveyed by both whole-genome and panel sequencing, colonized by mutant clones likewise depended on age ($p=0.002$) and severe chronic inflammation ($p=0.001$), but not metaplasia ($p=0.86$, ANOVA tests) (**Extended Data Fig. 9a-b**). While the age-dependency can be explained by the age-related increase in mutation burden, the effect of chronic inflammation on mutant proportion persists beyond differences in mutation burden (**Extended Data Fig. 9c**). On average, in 60-year-old individuals approximately 7.8% of glands were colonised by clones with driver mutations."

We have also added the following sentences to the discussion section:

"Severe chronic inflammation was significantly associated with elevated numbers of driver mutations in gastric glands and overall proportions of mutant epithelium in this study, highlighting a role for chronic inflammation in molding the pre-neoplastic selection landscape, as also identified in inflammatory bowel disease³⁰. Beyond a role for inflammation, the large variation in this proportion across donors may indicate between-donor variation in selective pressures. Larger studies may be powered to find exposures or other agents that further impose the selection of specific clones, as has been found for smoking in oesophagus⁴, and promote the transformation of normal cell to overt tumours via metaplasia and dysplasia."

2.15 Line 355. "This degree of similarity in mutational processes between segments of gastrointestinal tract is presumably a testament to the effectiveness of the various protective mechanisms operative between luminal contents and epithelial stem cells". This implicitly says that we would expect differences in mutational process exposure between the segments, could the authors please elaborate a bit on those?

Response: We thank the reviewer for highlighting this point, and have now spelled out the differences in luminal contents between the four segments, especially in regards to pH and microbiome:

"The somatic mutation landscapes of the four major segments of the gastrointestinal tract, the oesophagus^{4,28}, stomach, small intestine⁶ and large intestine⁵ have now been surveyed to a first level of resolution exhibiting illustrative similarities and differences. The lining epithelial cells of these segments interface with very different types of luminal content; these include air and the highly variable temperatures of food in the oesophagus, the acid and sterile contents of the stomach reservoir, the neutral contents and limited microbiome of the small intestine and the florid, diverse microbiome of the large intestine. Overall, however, the differences in somatic mutation rates and mutational signatures are modest."

2.16 Line 392. "In the stomach, the gland structure, and perhaps iterative damage and repair, may allow wider colonisation of the epithelial lining than in the small and large intestine." Could the authors please elaborate on what aspects of the gland structure of the stomach would allow wider colonisation than that of the intestines.

Response: We have clarified this section:

"These differences may, at least in part, reflect the epithelial architecture with the continuous stratified squamous epithelial sheet of the oesophagus allowing lateral spread of clones, whereas the crypt structure of the small intestine and large intestine hinders clones with drivers spreading beyond the confines of the individual crypt. In the stomach, the branching structure of pits into tubules, and perhaps iterative damage and repair, may allow wider colonisation of the epithelial lining than in the intestine."

2.17 Extended Data Figure 3/4. Please add all the gain timings to those if applicable and comment on them.

Response: We have added all gain timings to **Extended Data Figure 3** and **Extended Data Figure 4**, as well as an estimate of the age at which these gains occurred across donors in **Extended Data Figure 7**. We briefly allude to these in the text.

2.18 Telomere lengths. The authors did not but could look at telomere lengths across age, cancer vs. non-cancer, and how they relate to the CNVs, SNVs loads, presence of metaplasia etc.

Response: We thank the reviewer for this suggestion and now include an analysis of the telomere length per sample by using TelomereCat. Our results are shown in **Extended Data Figure 1f** and further described in the text:

“From whole-genome sequencing, we estimate that telomeres shorten by an average of 38 bases per year (95% confidence interval: 25-53) in gastric glands, with a significant shortening of telomeres by a mean of 570 bases in the presence of moderate or severe chronic inflammation ($p=6 \times 10^{-5}$, ANOVA test, **Extended Data Fig. 1f**). Beyond the effect of chronic inflammation, metaplasia did not further reduce telomere length ($p=0.11$, ANOVA test).”

2.19 Translocations, tandem duplications, inversions. Although the authors called them with GRIDSS, there is no mention of structural variants at all, just CNVs. Given that stomach glands are relatively unstable and present CNVs, it is not clear if copy-neutral structural changes were not detected at all or were not reported by the authors. It would be interesting to see the load and types of SVs (even if the numbers are 0's), and contrast them with the cancers.

Response: We thank the reviewer for this suggestion and now include a more detailed analysis of SVs beyond those supported as CNVs, resulting in calling more duplications, deletions and a few inversions, reported in **Fig. 4a, c**. Interestingly, the burden of intrachromosomal SVs is significantly higher in metaplastic than non-metaplastic glands, even after correcting for age. All SVs (besides those already called as CNVs) are now reported in **Extended Data Table 5**.

Referee #2 (Remarks on code availability):

The code looks ok but there is no README.

This is the second major comment in my comments to the authors:

Line 681. Code availability. The github code should have a README with a clear description, without which it is unnecessarily challenging to reproduce the analyses.

Response: This is addressed above.

Referee #3 (Remarks to the Author):

This study by Coorens et al reports about somatic mutation landscapes in normal and in cancerous stomach epithelial cells, by sequencing (usually clonal) microdissections of gastric glands. It is a largely rigorous study, however see comments to be addressed below; similarly so there are some concerns with reporting. The novelty does not lie in the general approach, which mirrors their previous work on other organs (e.g. colon, uterus). Instead, the strength of the study are some rather striking findings on existence of hypermutant glands (which are unevenly distributed) and the not-uncommon copy number alterations observed in healthy cells and their very intriguing recurrence within an individual. While these two findings are based on a low number of observations (in terms of individuals) and thus in a sense anecdotal, in my opinion they do provide an important basis for future more comprehensive research of these phenomena. Other findings of some interest include the change in mutation spectra between normal and cancerous cells (this was known for APOBEC), as well as in the driver mutation spectra between normal and cancerous cells (the CTNNB1 association is indeed, as they write, intriguing). Studying premalignant events may yield insights relevant for cancer early detection and prevention. I have some concerns about the analyses and reporting as listed below:

3.1 VAF distribution queries.

3.1a. In Fig 1b -- In 1 of the 3 shown patients, VAFs tend to be centered at $\sim 0.35-0.4$, quite consistently. What could be causing that?

Response: We thank the reviewer for highlighting this point. Lower VAF distributions can be caused by two related aspects: inadvertent inclusion of stromal/non-epithelial cells in the microdissection or a co-existence of secondary (small) clones within the epithelial cell populations, the latter of which the reviewer alludes to in the next point. As outlined in the response to the next point, we have now included a more in-depth analysis and discussion of the clonality of gastric glands.

3.1b. Related, VAF distributions of some (few) microdissections appear to result from a mix of clones. These could be easily identified by e.g. a mixture model or such. This might influence technical factors in bioinformatics pipelines. Do they present as outliers in the analysis associating mutation burden with age (or with H.pylori), or in the mutational signature analysis?

Response: We have included a more detailed description of the clonality of the gastric glands, as determined by their median VAF in **Figure 1c, Extended Data Figure 1a-b** and the accompanying text. Samples with low median VAF do present as outliers in the mutation burden analysis, nor the signature analysis. Metaplastic glands, however, have a significantly higher median VAF distribution than non-metaplastic glands.

“The clonal composition of gastric glands can be estimated from the variant allele fractions (VAFs) of somatic single nucleotide variants (SNVs) and small insertions and deletions (indels). The median VAF per microdissection generally exceeded 0.25 (**Fig. 1b,c**), which – allowing for a degree of stromal contamination – confirms the notion that most glands are dominated by the progeny of a single stem cell, a clone which takes up more than half of all cells. In 8% of microdissections (17/217), the median VAF was below 0.25, or had evidence of multiple clones co-existing, suggesting a continued presence of multiple stem cell niches and hierarchies more complicated than those observed in intestinal crypts⁵.”

As well as:

“Metaplastic glands exhibited overall higher median VAFs per microdissection compared to non-metaplastic glands ($p=0.006$, Wilcoxon rank sum test, **Extended Data Fig. 1b**), and were closely related phylogenetically when anatomically close to each other (**Extended Data Fig. 2a**), suggesting that metaplastic clones locally expand.”

3.2. Statistical consideration of how missing data affects interpretation of analyses.

3.2a. Lines 130-135. „Pathology review of hypermutant glands in the antrum revealed that six out of seven donors exhibited local chronic inflammation“ What proportion of non-hypermutant glands exhibited local chronic inflammation by the same criteria?

Response: We have now added pathology grading of chronic inflammation and intestinal metaplasia to all microdissected glands, and have revised the manuscript to focus on metaplastic glands rather than “hypermutant glands”, as suggested by Referee #1. The extent of chronic inflammation and metaplasia for all donors is now shown in **Fig. 2**. In the text, we now state:

“While chronic inflammation was widespread in both non-cancer and cancer donors, intestinal metaplasia, both complete and incomplete, was exclusive to the antrum of individuals with gastric cancer in our cohort.”

2b. Line 137. „Annotated current or previous *H. pylori* status, where known, did not significantly affect SNV burdens ($p=0.07$...“ $p=0.07$ probably favors that the association of Hpy to mutation rates exists rather than the converse, and I think this paragraph might be understood differently than they intended. If there really were past, undetected infections with effects on mutation rates, this association would be biased conservatively. Probably needs rewording. Also check association with individual mutational signatures, if not checked already.

Response: We have corrected one mix-up of *H. pylori* status for the two 64-year old male cancer samples, which has changed the statistical testing ($p=0.74$). However, only a minority of donors

have annotated *H. pylori* status (11/30). The only aberrant patterns of mutational signatures we find in our data concern the metaplastic samples.

We have revised this sentence:

“Annotated current or previous *H. pylori* status, known in only a minority of donors, did not significantly affect SNV burdens ($p=0.74$, ANOVA test, **Extended Data Fig. 1c**). However, the possibility of undetected infections affecting mutation rates precludes a definitive conclusion on this relationship.”

3.3. I have several worries about the mutational signature analysis:

3.3a. Lines 196-202 describe SBS mutsig in hypermutant glands. Increase in the ageing-associated SBS1 and SBS5/40 (1.7-4 fold) is smaller than the increase in non-ageing-associated (to my knowledge) SBS18 (11-fold), but they conclude the hypermutation is due to increased proliferation of stem cells? This does not quite fit. Present additional evidence and/or clarify, or remove „it is plausible“ claim.

Response: With the shift in focus on pathology annotation (metaplasia), we re-analysed the age-related burden for the three ubiquitous mutational signatures (SBS1, SBS5/40 and SBS18). It is now significantly associated with age ($p=0.047$, **Extended Data Fig. 5f**). We have updated the text accordingly. The clock-like behaviour for SBS18 has also been reported in the colon (Lee-Six et al., 2019, *Nature*) and small intestine (Wang et al., 2023, *Nat. Genet.*). Note that this response is identical to point **2.10** above.

3.3b. Regarding ID1/ID2 ratio being significantly changed in hypermutating glands – again it is not clear how this links to increased proliferation (lines 210-212). Is ID2 known to be more strongly proliferation associated with ID1 – where is the data/reference to support that? In absence thereof the claim does not appear to stand.

Response: As this revision included more rigorous indel signature analysis (see answer to point **3.3c**), we have revised this section, and moved the interpretation of signature results to the discussion. Specifically, this section now reads:

“The excess of indels observed in metaplastic glands was primarily composed of ID1 and ID2 (**Fig. 3f; Extended Data Fig. 6**) but the ratio of ID2 to ID1 was significantly elevated in metaplastic glands compared to other non-cancer glands and was more akin to that of gastric cancer samples (**Fig. 3g**).”

In the Discussion:

Gastric glands with intestinal metaplasia, which are often associated with chronic inflammation and local clonal expansions, show increases in total mutation burdens due to elevated SBS1 (methylcytosine deamination), SBS18 (reactive oxygen species), ID1 and ID2 (replication strand

slippage) and intrachromosomal SV mutation rates. These changes in mutagenesis could reflect an increase in cell division rates in metaplastic glands, be the consequence of other factors intrinsic to such cells or be due to microenvironmental influences.

3.3c. Line 630: „As ID1 and ID2 are highly specific signatures, their contribution was estimated based on the number of single base insertions and deletions of homopolymer runs of A and T of length six or greater, respectively.“ Justify why for indels the same procedure was not used as for SNVs? (based on mSigHdp + postprocessing).

Response: We have now extracted indel signatures in the same way as the SNV signatures, using the HDP R package (<https://github.com/nicolaroberts/hdp>). The results are plotted in **Extended Data Fig. 6** and **Fig. 3f-g**, described in the text, and indel signature proportions are included in **Extended Data Table 2**. We extracted five indel HDP signatures, which were deconvolved into COSMIC reference signatures ID1, ID2, ID5, ID6, ID9, ID12 and ID14. HDP indel signature 5, a noisy signature typified by large deletions, was not decomposed further as no combination of reference signatures yielded a high enough cosine similarity to the extracted signature.

“Signature analysis of indels revealed that gastric glands exhibited ID1 and ID2, single base insertions and deletions respectively at homopolymer runs of T/A linked to polymerase slippage, and ID5 and ID9, both characterized by deletions at homopolymers but of unknown aetiology. The excess of indels observed in metaplastic glands was primarily composed of ID1 and ID2 (**Fig. 3f; Extended Data Fig. 6**) but the ratio of ID2 to ID1 was significantly elevated in metaplastic glands compared to other non-cancer glands and was more akin to that of gastric cancer samples (**Fig. 3g**).”

3.3d. Line 226. „The pattern of mutational signatures in glands dissected from the two gastric cancers was markedly different from that in normal glands.“ Please state the proportion of signatures in the text (ideally do a statistical test, although sample size might preclude this). APOBEC contribution does not seem major (in line with APOBEC mutagenesis not being that highly prevalent in stomach cancers).

Response: Given that only two gastric cancers are included in this study, we use them as illustrative examples for comparison rather than a rigorous and widely sampled set to test the normal microdissections against, especially as some processes are (nearly) exclusive to cancers (ID6/SBS3; SBS17a/17b; SBS2/SBS13). The proportions of SNVs signatures in the two gastric cancers are visually represented in **Fig. 3c-d** and are included in **Extended Data Table 2** (both indels and SNVs).

3.4. Some concerns regarding driver alterations

3.4a. They seem to comment only on the SNV in this context, but not on the CNV. Are the 3 types of CNVs (focal, arm level ccn-loh, and whole-chromosome events) statistically enriched with known dosage-sensitive oncogenes, or TSGs? (that there is an enrichment in fragile sites does not preclude there is also an enrichment in driver events) Do they observe two-hit events with the CNV, e.g. possibly some germline variants might cause selection on the recurrent CNV within 1 individual?

Response: Recurrent CNVs within one individual usually comprise trisomies, in which the gained copy is evenly distributed across maternal/paternal copies. This essentially rules out that the amplification of a specific germline variant is underlying this phenomenon and suggests a possible selection for a dosage effect. However, determining the precise gene or genes that drive this dosage effect is not possible within the confines of this data. As the reviewer points out, the other recurrent structural variants occur at common fragile sites and usually affect the listed gene. While these structural variants at fragile sites have been identified before, their nature precludes a definitive answer as to whether the high rate is due to background mutagenesis, a read-out of wider genomic instability, or positive selection.

We have added this sentence to the text:

“Intrachromosomal CNVs/SVs were predominantly deletions and many involved well-known fragile sites in *FHIT*, *PTPRD*, and *MACROD2*^{23,24}. While these have been reported in gastric cancer²³, it is unclear whether the observed events are due to high rates of mutagenesis in the locus or positive selection in this data.”

No CNVs/SVs overlap the identified driver mutations, so there is observation of a somatic two-hit phenomenon in this data set.

3.4b. Driver mutations and age „The average number of driver mutations per gland per individual correlated with age ($p=0.008$ “ does this association remain when controlling for general increase of mutations with age?

Response: Upon the suggestion of the reviewer, we have now included an additional analysis to assess the relation between aging, accumulation of mutations and driver preponderance. While generally the proportion of driver mutations and age or mutation burden, we find that severe chronic inflammation increases the proportion of drivers beyond what is expected by age or mutation burden.

The text now reads:

“The proportion of the gastric epithelium, surveyed by both whole-genome and panel sequencing, colonized by mutant clones likewise depended on age ($p=0.002$) and severe chronic inflammation ($p=0.001$), but not metaplasia ($p=0.86$, ANOVA tests) (**Extended Data Fig. 9a-b**). While the age-dependency can be explained by the age-related increase in mutation burden, the

effect of chronic inflammation on mutant proportion persists beyond differences in mutation burden (**Extended Data Fig. 9c**). On average, in 60-year-old individuals approximately 7.8% of glands were colonised by clones with driver mutations.”

3.4c. Potentially important concern about method to detect driver variants. Line 647: „We used the dndscv24 R package to identify genes under positive selection, combining both the whole-genome sequencing data and the targeted sequencing data“ It is not clear how the 2 data types can be combined here: the modelling of mutation risk from covariates in dNdScv would not be expected to work from targeted sequencing data. Can they explain better how did they go about this? Can they demonstrate with data that dNdScv is safe to use in this way (mixing WES and panel sequencing, if they did so)?

Response: We thank the reviewer for highlighting this point and prompting a closer look at the analysis. The covariates used by dNdScv to model the background mutation rate are supplied to the algorithm rather than inferred from the data itself (Martincorena et al., 2017, *Cell*). The difference in coverage between genes included in the targeted panel and those that are not is internally handled per gene, as this will influence the overall detection of mutations, both synonymous and non-synonymous. The inference of dN/dS ratios and driver discovery from dNdScv has been applied to targeted sequencing data (see Martincorena et al., 2015, *Science* and Martincorena et al., 2018, *Science*), as well as mixed WES/WGS and targeted sequencing data (see Lawson et al., 2020, *Science*), provided no mutations are double-counted if WES/WGS and targeted sequencing data are combined from the same donor.

However, when doing this, it is vital to split the analysis for genes inside the targeted panel and those outside, which we did not do previously, but have now implemented. This reanalysis of the genes inside the panel (with data from both WGS and targeted sequencing) resulted in the loss of *ERBB3* and the inclusion of *ARID2* and *EEF1A1* as genes under significant positive selection. The analysis of the WGS data alone did not reveal any additional genes under positive selection.

We have revised **Fig. 5** and updated the **Methods** section accordingly.

3.5. Bioinformatics question. Line 578: „The truncated distribution is necessary to reflect the minimum number of reads that support a variant ($n = 4$) that is imposed by variant callers such CaVEMan“ My intuition is that forcing 4 reads to support a variant may be quite stringent, and aggravates the confounding by sequencing depth, which then necessitates they have to correct for it (lines 591-597) introducing extra complexity and possibly errors.

Response: The rate of false positives increases with lower read support, such that allowing variants only supported by two or three reads will likely have a higher proportion of artefactual variants. As most microdissections of single glands have a high median VAF and the median

coverage is 23, the general sensitivity (included in **Extended Data Table 2**) is high and generally exceeds 0.8 (197/217, 90% of microdissections) and the majority is above 0.95 (134/217, 61%). While stringent, a cut-off of 4 reads will enable the cleanest downstream analyses, including signatures and phylogenies.

*Histogram of the estimated sensitivity of SNV calling based on median VAFs and coverage. Data can be found in **Extended Data Table 2**.*

3.6. Data availability. Please make the somatic mutation calls available (not only raw data/reads), preferably without restrictions, to facilitate reproducibility and reuse.

Response: We have now deposited all the filtered somatic mutation calls, for indels and SNVs, per donor into GitHub. Since unfiltered calls would include germline variants, we cannot make these publicly available as these could lead to donor identification.

3.7. Methods text could be improved by edits. Below are suggestions for some clarifications/corrections/added references. There could be others; please go over methods with a fine-tooth comb and correct errors and ambiguities and citations.

Line 575-576 „For every indel or SNV, the overdispersion parameter (ρ) was determined in a grid-based way (ranging the value of ρ from 10^{-6} to $10^{-0.05}$)“ is confusing/incomplete. „Grid-based“ but only one parameter is varied (not two or more)? What is the criteria for choosing best ρ ?

Response: We have clarified the language in this section - the value of ρ is estimated by maximum likelihood out of the discretized set of values for ρ (ranging the value of $\log_{10}(\rho)$ from -6 to -0.05 in steps of 0.05).

Line 602: „we included site-specific age relations in the mixed effects model“ vague – I infer this is an interaction term age:site? Please include all details necessary to understand and reproduce analysis.

Response: We have clarified the text to make sure it is clear we used an interaction term age:site, as intuited by the reviewer. Of note, all scripts used for analysis, including the linear mixed effects modelling are now available on the accompanying Github repo (<https://github.com/TimCoorens/stomach>).

For this specific analysis:

“To test the effect of gastric site on the mutation rate, we included site-specific age relations in the mixed effects model.

1: Burden ~ Age:Site + IM + (1 | Donor)”

Line 611: „To identify possibly undiscovered mutational signatures in human placenta“ Placenta? Are the other parts of the Method also copy-pasted from another text without checking well if they apply here? Please check.

Response: We apologise for the oversight and confirm that we have gone through the entire text multiple times to ensure no rogue references to other data sets remain.

3.8. Further minor remarks.

SBS17 signature consists of A>C (T>G) mutations, which do have reported links with acid exposure (in a model system) and oxidation of the free nucleotide pool. This seems a more relevant explanation than the 5-FU association of SBS17 that is mentioned.

Response: We have added this possible interpretation of SBS17 to its description:

“SBS17a and SBS17b, of unknown aetiologies but sometimes associated with exposure to the chemotherapeutic agent 5-fluorouracil or oxidation of the free nucleotide pool²⁰”

Line 165: Does SBS18 have to result from *endogenous* ROS?

Response: We have removed the “endogenously generated” from this sentence.

Line 567-569: „Germline variants were removed using a one-sided binomial exact test on the number of variant reads and depth present across largely diploid samples, as previously described.“ Seems to lack a reference.

Response: We have fixed the lacking reference.

Line 523: „Custom Agilent SureSelect bait set capturing the exonic regions of the following 321 cancer associated genes:“ please put them in a supp table instead of listing them here, which is distracting.

Response: The set of cancer-associated genes within the bait set is now listed in **Extended Data Table 7**.

Discussion is quite long.

Response: We have tidied up and reduced the length of the Discussion where we feel able. While still sizeable, it puts the genomic findings from this study in a larger context with similar studies on the gastrointestinal tract.